# Pituitary stem cells produce paracrine WNT signals to control the expansion of their descendant progenitor cells

John P Russell[1], Xinhong Lim[2,3], Alice Santambrogio[1,4], Val Yianni[1], Yasmine Kemkem[5], Bruce Wang[6,7], Matthew Fish[6], Scott Haston[8], Anaëlle Grabek[9], Shirleen Hallang[1], Emily J Lodge[1], Amanda L Patist[1], Andreas Schedl[9], Patrice Mollard[5], Roel Nusse[6], Cynthia L Andoniadou[1,4]*

[1]Centre for Craniofacial and Regenerative Biology, King's College London, London, United Kingdom; [2]Skin Research Institute of Singapore, Agency for Science, Technology and Research, Singapore, Singapore; [3]Lee Kong Chian School of Medicine, Nanyang Technological University, Singapore, Singapore; [4]Department of Medicine III, University Hospital Carl Gustav Carus, Technische Universität Dresden, Dresden, Germany; [5]Institute of Functional Genomics (IGF), University of Montpellier, CNRS, Montpellier, France; [6]Howard Hughes Medical Institute, Stanford University School of Medicine, Department of Developmental Biology, Stanford University School of Medicine, Stanford, United States; [7]Department of Medicine and Liver Center, University of California San Francisco, San Francisco, United States; [8]Developmental Biology and Cancer, Birth Defects Research Centre, UCL GOS Institute of Child Health, London, United Kingdom; [9]Université Côte d'Azur, Inserm, CNRS, Nice, France

**Abstract** In response to physiological demand, the pituitary gland generates new hormone-secreting cells from committed progenitor cells throughout life. It remains unclear to what extent pituitary stem cells (PSCs), which uniquely express SOX2, contribute to pituitary growth and renewal. Moreover, neither the signals that drive proliferation nor their sources have been elucidated. We have used genetic approaches in the mouse, showing that the WNT pathway is essential for proliferation of all lineages in the gland. We reveal that SOX2[+] stem cells are a key source of WNT ligands. By blocking secretion of WNTs from SOX2[+] PSCs in vivo, we demonstrate that proliferation of neighbouring committed progenitor cells declines, demonstrating that progenitor multiplication depends on the paracrine WNT secretion from SOX2[+] PSCs. Our results indicate that stem cells can hold additional roles in tissue expansion and homeostasis, acting as paracrine signalling centres to coordinate the proliferation of neighbouring cells.

*For correspondence:
cynthia.andoniadou@kcl.ac.uk

## Introduction

How stem cells interact with their surrounding tissue has been a topic of investigation since the concept of the stem cell niche was first proposed (*Schofield, 1978*). Secreted from supporting cells, factors such as WNTs, FGFs, SHH, EGF, and cytokines regulate the activity of stem cells (*Nabhan et al., 2018*; *Palma et al., 2005*; *Tan and Barker, 2014*). Furthermore, communication is known to take place in a bidirectional manner (*Doupé et al., 2018*; *Tata and Rajagopal, 2016*).

The anterior pituitary (AP) is a major primary endocrine organ that controls key physiological functions including growth, metabolism, reproduction, and the stress response and exhibits tremendous capability to remodel its constituent hormone populations throughout life, in response to

physiological demand. It contains a population of *Sox2* expressing stem cells that self-renew and give rise to lineage-committed progenitors and functional endocrine cells (*Andoniadou et al., 2013*; *Rizzoti et al., 2013*). During embryonic development, SOX2$^+$ undifferentiated precursor cells of Rathke's pouch, the pituitary anlage (*Arnold et al., 2011*; *Castinetti et al., 2011*; *Fauquier et al., 2008*; *Pevny and Rao, 2003*), generate all committed endocrine progenitor lineages, defined by the absence of SOX2 and expression of either POU1F1 (PIT1), TBX19 (TPIT), or NR5A1 (SF1) (*Bilodeau et al., 2009*; *Davis et al., 2011*). These committed progenitors are proliferative and give rise to the hormone-secreting cells. Demand for hormone secretion rises after birth, resulting in dramatic organ growth and expansion of all populations by the second postnatal week (*Carbajo-Pérez and Watanabe, 1990*; *Taniguchi et al., 2002*). SOX2$^+$ pituitary stem cells (PSCs) are most active during this period, but the bulk of proliferation and organ expansion during postnatal stages derives from SOX2$^-$ committed progenitors. The activity of SOX2$^+$ PSCs gradually decreases and during adulthood is minimally activated even following physiological challenge (*Andoniadou et al., 2013*; *Gaston-Massuet et al., 2011*; *Gremeaux et al., 2012*; *Zhu et al., 2015*). By adulthood, progenitors carry out most of the homeostatic functions, yet SOX2$^+$ PSCs persist throughout life in both mice and humans (*Gonzalez-Meljem et al., 2017*; *Xekouki et al., 2019*). The signals driving proliferation of committed progenitor cells are not known, and neither is it known if SOX2$^+$ PSCs can influence this process beyond their minor contribution of new cells.

The self-renewal and proliferation of numerous stem cell populations rely on WNT signals (*Basham et al., 2019*; *Lim et al., 2013*; *Takase and Nusse, 2016*; *Wang et al., 2015*; *Yan et al., 2017*). WNTs are necessary for the initial expansion of Rathke's pouch as well as for PIT1 lineage specification (*Osmundsen et al., 2017*; *Potok et al., 2008*). In the postnatal pituitary, the expression of WNT pathway components is upregulated during periods of expansion and remodelling. Gene expression comparisons between neonatal and adult pituitaries or in GH-cell ablation experiments (*Gremeaux et al., 2012*; *Willems et al., 2016*) show that the WNT pathway is upregulated during growth and regeneration.

Our previous work revealed that during disease, the paradigm of supporting cells signalling to the stem cells may be reversed; mutant stem cells expressing a degradation-resistant β-catenin in the pituitary promote cell non-autonomous development of tumours through their paracrine actions (*Andoniadou et al., 2013*; *Gonzalez-Meljem et al., 2017*). Similarly, degradation-resistant β-catenin expression in hair follicle stem cells led to cell non-autonomous WNT activation in neighbouring cells promoting new growth (*Deschene et al., 2014*). In the context of normal homeostasis, stem cells have been shown to influence daughter cell fate in the mammalian airway epithelium and the *Drosophila* gut via 'forward regulation' models, where the fate of a daughter cell is directed by a stem cell via juxtacrine Notch signalling (*Ohlstein and Spradling, 2007*; *Pardo-Saganta et al., 2015*). It remains unknown if paracrine stem cell action can also promote local proliferation in normal tissues.

Here, we used genetic approaches to determine if paracrine stem cell action takes place in the AP and to discern the function of WNTs in pituitary growth. Our results demonstrate that postnatal pituitary expansion, largely driven by committed progenitor cells, depends on WNT activation. Importantly, we show that SOX2$^+$ PSCs are the key regulators of this process, acting through secretion of WNT ligands acting in a paracrine manner on neighbouring progenitors. Identification of this forward-regulatory model elucidates a previously unidentified function for stem cells during tissue expansion.

## Results

### WNT-responsive cells in the pituitary include progenitors driving major postnatal expansion

To clarify which cells respond to WNT signals in the postnatal AP, we first characterised the AP cell types activating the WNT pathway at P14, a peak time for organ expansion and a time point when a subpopulation of SOX2$^+$ stem cells are proliferative. The *Axin2-CreERT2* mouse line (*van Amerongen et al., 2012*) has been shown to efficiently label cells with activated WNT signalling in the liver, lung, breast, skin, testes, and endometrium among other tissues (*Lim et al., 2013*; *Moiseenko et al., 2017*; *Syed et al., 2020*; *van Amerongen et al., 2012*; *Wang et al., 2015*). *Axin2* positive cells were labelled by GFP following tamoxifen induction in *Axin2$^{CreERT2/+}$*;

*ROSA26^{mTmG/+}* mice and pituitaries were analysed 2 days post-induction. We carried out double immunofluorescence staining using antibodies against uncommitted (SOX2), lineage committed (PIT1, TPIT, SF1), and hormone-expressing endocrine cells (GH, PRL, TSH, ACTH, or FSH/LH) together with antibodies against GFP labelling the WNT-activated cells. We detected WNT-responsive cells among all the different cell types of the AP including SOX2$^+$ PSCs, the three committed populations and all hormone-secreting cells (*Figure 1A*, *Figure 1—figure supplement 1A*).

To confirm if the three committed lineages as well as uncommitted SOX2$^+$ PSCs all expand in response to WNT, we further lineage traced *Axin2*-expressing cells for 14 days after tamoxifen administration at P14. Double labelling revealed an increase in all four populations between 2 and 14 days (*Figure 1A,B*). This increase reached significance for the PIT1 (13.7% at 2 days to 30.3% at 14 days, p=0.000004) and TPIT (3.78% to 11.03%, p=0.008) populations, but not SF1 (0.5% to 4%, n.s.). As this time course ends at P28 at the commencement of puberty, we repeated the analysis for SF1 cells to P42, which spans puberty and the expansion of gonadotrophs (*Figure 1—figure supplement 1B*). This reveals a significant expansion in WNT-responsive SF1$^+$ cells as a proportion of the total SF1$^+$ population (p=0.0048, n = 3). Lineage tracing of the PIT1-derivates (GH$^+$ somatotrophs, PRL$^+$ lactotrophs, and TSH$^+$ thyrotrophs) reveals that there is a preferential expansion of somatotrophs and thyrotrophs (*Figure 1—figure supplement 1C*). Only a minority of SOX2$^+$ PSCs were WNT-responsive at 2 days (0.57%) and this population expanded to 2% at 14 days (n.s.), suggesting that these are self-renewing. GFP$^+$ cells were traced for a period of 8 weeks post-induction, which revealed that WNT-responsive descendants continued to expand at the same rate as the rest of the pituitary (n = 4–8 mice per time point at P16, P21, P28, P42, and P70) (*Figure 1C,D*). The time period between 2 and 7 days saw the greatest increase in GFP$^+$ cells, during which the labelled population nearly tripled in size (*Figure 1D*). The persistence of labelled cells was evident in longer-term traces using the *ROSA26^{lacZ/+}* reporter (*Axin2^{CreERT2/+};ROSA26^{lacZ/+}*), up to a year following induction at P14 (*Figure 1E*, n = 4). Clonal analysis using the Confetti reporter demonstrated that individual *Axin2*-expressing cells (*Axin2^{CreERT2/+};ROSA26^{Confetti/+}*) gave a greater contribution after 4 weeks compared to lineage tracing from *Sox2*-expressing cells (*Sox2^{CreERT2/+};ROSA26^{Confetti/+}*), in support of predominant expansion from WNT-responsive lineage-committed progenitors (*Figure 1—figure supplement 1D*).

To establish if signalling mediated by β-catenin is necessary for organ expansion we carried out deletion of *Ctnnb1* in the *Axin2$^+$* population from P14 during normal growth (*Axin2^{CreERT2/+}; Ctnnb1^{lox(ex2-6)/lox(ex2-6)}* hereby *Axin2^{CreERT2/+};Ctnnb1^{LOF/LOF}*). Due to morbidity, likely due to *Axin2* expression in other organs, we were limited to analysis up to 5 days post-induction. Deletion of *Ctnnb1* resulted in a significant reduction in the number of dividing cells, marked by pH-H3 (40% reduction, *Figure 1—figure supplement 2A*, p=0.0313, n = 3), confirming that activation of the WNT pathway is necessary for expansion of the pituitary populations. This deletion did not result in significant differences in overall numbers among the three lineages, as determined by the numbers of PIT1$^+$, SF1$^+$, or ACTH$^+$ cells among the targeted population (*Figure 1—figure supplement 2B*, n = 4 controls, two mutants). The number of SOX2$^+$ stem cells and cells undergoing cell death also remained unaffected during the 5-day period (*Figure 1—figure supplement 2C and D*). Taken together, these results confirm that postnatal AP expansion depends on WNT-responsive progenitors across all lineages, in addition to SOX2$^+$ PSCs (*Figure 1F*).

## WNT/β-catenin signalling is required for long-term AP expansion from SOX2$^+$ PSCs

We further explored the role of WNT pathway activation in postnatal SOX2$^+$ stem cells. To permanently mark WNT-responsive cells and their descendants whilst simultaneously marking SOX2$^+$ PSCs, we combined the tamoxifen-inducible *Axin2^{CreERT2/+};ROSA26^{tdTomato/+}* with the *Sox2^{Egfp/+}* strain, where cells expressing SOX2 are labelled by enhanced green fluorescent protein (EGFP) (*Axin2^{CreERT2/+};Sox2^{Egfp/+};ROSA26^{tdTomato/+}*). Following tamoxifen administration from P21, tdTomato- and EGFP-labelled cells were analysed by flow sorting after 72 hr, by which point all induced cells robustly express tdTomato (*Figure 2A*, *Figure 2—figure supplement 1*). Double-labelled cells comprised 23.4% of the SOX2$^+$ population (n = 5 individual pituitaries) (*Figure 2A*, arrowheads), with the majority of tdTomato$^+$ cells found outside of the SOX2$^+$ compartment. It was previously shown that only around 2.5–5% of SOX2$^+$ PSCs has clonogenic potential through in vitro assays (*Andoniadou et al., 2012*; *Andoniadou et al., 2013*; *Pérez Millán et al., 2016*). To determine if

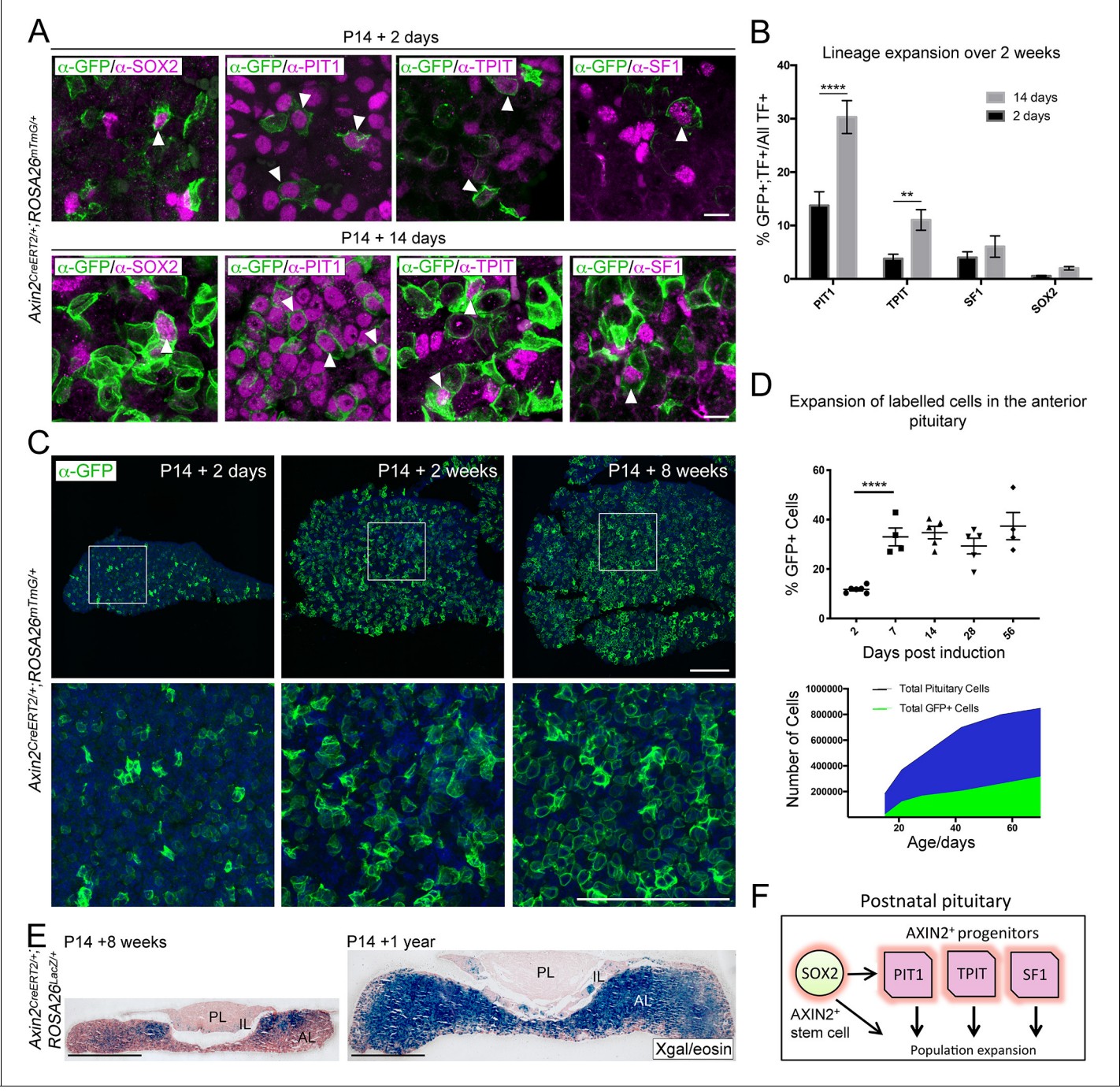

**Figure 1.** *Axin2* expressing cells contribute to pituitary growth and expansion of all lineages. (**A**) Immunofluorescence staining against GFP (green) with markers of pituitary stem cells (PSCs) or lineage commitment (magenta) in *Axin2*$^{CreERT2/+}$; *ROSA26*$^{mTmG/+}$ pituitaries harvested from mice induced at P14 and lineage traced for 2 days (top panel) and 14 days (bottom panel). Scale bar: 10 μm. (**B**) Quantification of lineage expansion between 2 and 14 days following induction at P14. Graph shows that the proportion of lineage committed cells (either PIT1$^+$, TPIT$^+$, or SF1$^+$) and PSCs (SOX2$^+$), that is, that are transcription factor (TF)$^+$ cells that are GFP$^+$ increases between 2 days (black bars) and 14 days (grey bars) post-induction. PIT1 p=0.000004, TPIT p=0.008 multiple *t*-tests. *n* = 4 animals per time point. (**C**) Immunofluorescence staining against GFP (green) in pituitaries harvested from *Axin2*$^{CreERT2/+}$;*ROSA26*$^{mTmG/+}$ mice induced at P14 and lineage traced for 2 days, 2 weeks, and 8 weeks. Bottom panel shows magnified fields of view of regions of interest indicated by white boxes in panels above. Scale bars: 50 μm. (**D**) Top panel showing the quantification of the proportion of all cells in *Axin2*$^{CreERT2/+}$;*ROSA26*$^{mTmG/+}$ pituitaries that are GFP$^+$ at 2, 7, 14, 28, and 56 days post-induction as analysed by flow cytometry. Days 2–7 p<0.0001 unpaired *t*-test. Data points show individual measurements from biological replicates, *n* = 4–8 pituitaries per time point. (Bottom) Graph of the absolute number of GFP+ cells (green) and as a proportion of total cells (blue) at the time points indicated. (**E**) X-gal staining in *Axin2*$^{CreERT2/+}$;

*Figure 1 continued on next page*

*Figure 1 continued*

ROSA26$^{LacZ/+}$ pituitaries harvested from mice induced at P14 and lineage traced for 8 weeks (left) and 1 year (right). Scale bars: 500 µm. (**F**) Model summarising the major contribution of WNT-responsive progenitors of all lineages to pituitary growth, in addition to that of SOX2$^+$ PSCs.

The online version of this article includes the following figure supplement(s) for figure 1:

**Figure supplement 1.** *Axin2* expressing cells contribute to pituitary growth and expansion of all lineages.

**Figure supplement 2.** *Axin2* expressing cells contribute to pituitary growth and expansion of all lineages.

WNT-responsive SOX2$^+$ cells are stem cells capable of forming colonies, we isolated double-positive tdTomato$^+$;EGFP$^+$ cells (i.e. *Axin2*$^+$;*Sox2*$^+$) as well as the single-expressing populations and plated these in equal numbers in stem cell-promoting media at clonal densities (*Figure 2B*). Double-positive tdTomato$^+$;EGFP$^+$ cells showed a significant increase in the efficiency of colony formation compared to single-labelled EGFP$^+$ cells (average 9% compared to 5%, *n* = 5 pituitaries, p=0.0226, Mann–Whitney *U*-test [two-tailed]), demonstrating WNT-responsive SOX2$^+$ PSCs have a greater clonogenic potential under these in vitro conditions, confirming in vivo data in *Figure 1B*. As expected from previous work, none of the single-labelled tdTomato$^+$ cells (i.e. SOX2 negative) was able to form colonies (*Andoniadou et al., 2012*).

To confirm that PSCs with active WNT signalling through β-catenin have a greater propensity to form colonies in vitro, we analysed postnatal pituitaries from TCF/Lef:H2B-EGFP mice, reporting the activation of response to WNT signals. This response is detected through expression of an EGFP-tagged variant of histone H2B, which is incorporated into chromatin and diluted in descendants with cell division (*Ferrer-Vaquer et al., 2010*). Therefore, cells responding to, or having recently responded to, WNT as well as their immediate descendants will be EGFP$^+$. At P21, EGFP$^+$ cells were abundant in all three lobes and particularly in the marginal zone harbouring SOX2$^+$ stem cells (*Figure 2—figure supplement 2A*). Through double mRNA in situ hybridisation against *Egfp* and *Sox2* in TCF/Lef:H2B-EGFP pituitaries, we confirmed that *Sox2*-expressing cells activate H2B-EGFP expression at this time point (*Figure 2—figure supplement 2B*). Isolation by fluorescence-activated cell sorting (FACS) and in vitro culture of the postnatal EGFP$^+$ compartment revealed an enrichment of cells with clonogenic potential in the EGFP$^{High}$ fraction compared to EGFP$^{Low}$ or negative cells (*Figure 2—figure supplement 2C*, *n* = 5 pituitaries). Together these results reveal that a proportion of postnatal SOX2$^+$ stem cells respond to WNTs through downstream β-catenin/TCF/LEF signalling and that these cells have greater clonogenic capacity in vitro.

To further address the role of the canonical WNT response in the activity of SOX2$^+$ PSCs in vivo, we expressed a loss-of-function allele of β-catenin specifically in *Sox2*-expressing cells (*Sox2*$^{CreERT2/+}$; *Ctnnb1*$^{lox(ex2-6)/lox(ex2-6)}$ hereby *Sox2*$^{CreERT2/+}$;*Ctnnb1*$^{LOF/LOF}$) from P14. Twenty-two weeks following induction, at P168, there was a substantial drop in the number of cycling cells in the pituitary of *Sox2*$^{CreERT2/+}$;*Ctnnb1*$^{LOF/LOF}$ mutants compared to *Sox2*$^{+/+}$;*Ctnnb1*$^{LOF/LOF}$ controls (*Figure 2C*, *n* = 2 pituitaries per genotype). This was accompanied by AP hypoplasia following the loss of *Ctnnb1* in SOX2$^+$ PSCs (*Figure 2D*). Therefore, in this small sample size, the proliferative capacity of *Ctnnb1*-deficient SOX2$^+$ PSCs and of their descendants was impaired long term, leading to reduced growth. In vivo genetic tracing of targeted cells over the 22-week period (*Sox2*$^{CreERT2/+}$;*Ctnnb1*$^{LOF/+}$; *ROSA26*$^{mTmG/+}$ compared to *Sox2*$^{CreERT2/+}$;*Ctnnb1*$^{LOF/LOF}$;*ROSA26*$^{mTmG/+}$ pituitaries) revealed that targeted (*Ctnnb1*-deficient) SOX2$^+$ PSCs were capable of giving rise to the three committed lineages PIT1, TPIT, and SF1 (*Figure 2—figure supplement 2D*), indicating that the loss of *Ctnnb1* does not prevent differentiation of SOX2$^+$ PSCs into the three lineages. Downregulation of β-catenin was confirmed by immunofluorescence in SOX2$^+$ (mGFP$^+$) derivatives (*Figure 2—figure supplement 2E*). Although limited by a small sample size, we conclude that WNT/β-catenin signalling is likely required cell-autonomously in SOX2$^+$ stem cells and their descendants (*Figure 2E*).

## SOX2$^+$ stem cells express WNT ligands

Having established that WNT activation is responsible for promoting proliferation in the AP, we next focused on identifying the source of WNT ligands. *Axin2* expressing cells from *Axin2*$^{CreERT2/+}$; *ROSA26*$^{mTmG/+}$ mice were labelled at P14 by tamoxifen induction. Cells expressing *Axin2* at the time of induction are labelled by GFP expression in the membrane. Double immunofluorescence staining for GFP together with SOX2 revealed that *Axin2* expressing cells (mGFP$^+$) are frequently

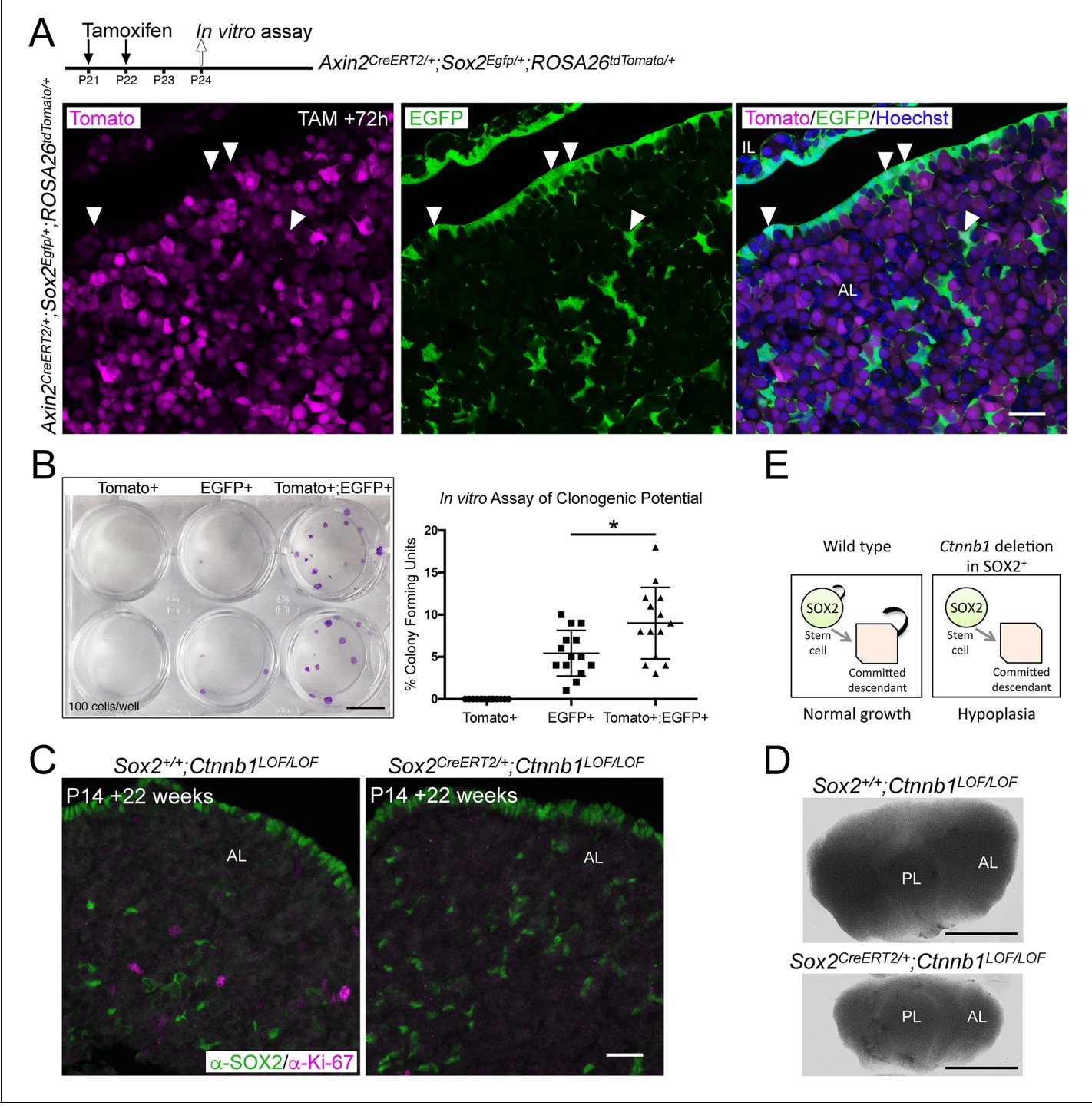

**Figure 2.** Activation of WNT signalling in SOX2[+] pituitary stem cells (PSCs) and their descendants is necessary for long-term growth. (**A**) Schematic of the experimental timeline used in panels **A** and **B**. Endogenous expression of tdTomato (magenta, *Axin2* targeted cells) and EGFP (green, *Sox2* expressing cells) in *Axin2^CreERT2/+^;Sox2^Egfp/+^;ROSA26^tdTomato/+^* pituitaries harvested at P24 sectioned in the frontal plane. Nuclei are counterstained with Hoechst in the merged panel. Scale bar: 50 µm. (**B**) A representative culture plate showing colonies derived from Tomato[+], EGFP[+], or Tomato[+];EGFP[+] cells that were isolated from *Axin2^CreERT2/+^;Sox2^Egfp/+^;ROSA26^tdTomato/+^* pituitaries by fluorescence-activated cell sorting (FACS) plated in stem cell promoting media at clonogenic densities and stained with crystal violet (left panel). The proportion of colony-forming cells in each subpopulation was quantified by counting the number of colonies per well (right panel). Each data point indicates individual wells, *n* = 5 separate pituitaries. p=0.0226, Mann–Whitney *U*-test (two-tailed). Scale bar: 10 mm. (**C**) Immunofluorescence staining against SOX2 (green) and Ki-67 (magenta) in *Sox2^+/+^Ctnnb1^LOF/LOF^* (control) and *Sox2^CreERT2/+^Ctnnb1^LOF/LOF^* (mutant) pituitaries from mice induced at P14 and analysed 22 weeks after induction (at P168) (bottom

*Figure 2 continued on next page*

*Figure 2 continued*

panel). Scale bar: 50 µm. (D) Dorsal view of whole mount pituitaries of *Sox2^{+/+};Ctnnb1^{LOF/LOF}* (control) and *Sox2^{CreERT2/+};Ctnnb1^{LOF/LOF}* (mutant), 22 weeks after induction (i.e. P168). Scale bars: 1 mm. (E) Model summarising the effect of *Ctnnb1* deletion in SOX2^+ PSCs. PL, posterior lobe; IL, intermediate lobe; AL, anterior lobe.

The online version of this article includes the following figure supplement(s) for figure 2:

**Figure supplement 1.** Activation of WNT signalling in SOX2^+ pituitary stem cells (PSCs) and their descendants is necessary for long-term growth.
**Figure supplement 2.** Activation of WNT signalling in SOX2^+ pituitary stem cells (PSCs) and their descendants is necessary for long-term growth.

located in close proximity to SOX2^+ PSCs (*Figure 3A*). Two-dimensional quantification of the two cell types revealed that over 50% of mGFP^+ cells were in direct contact with SOX2^+ nuclei (*n* = 3 pituitaries, >500 SOX2^+ cells per gland, *Figure 3A*). The analysis did not take into account the cellular processes of SOX2^+ cells. These results led us to speculate that SOX2^+ PSCs may be a source of key WNT ligands promoting proliferation of lineage-committed cells.

In order to determine if SOX2^+ PSCs express WNT ligands, we carried out gene expression profiling of SOX2^+ and SOX2^− populations at P14, through bulk RNA-sequencing. Pure populations of *Sox2*-expressing cells excluding lineage-committed populations were isolated from *Sox2^{Egfp/+}* male and female pituitaries at P14 based on EGFP expression as previously shown (*Andoniadou et al., 2012*; *Figure 3B*, *Figure 3—figure supplement 1A*). Analysis of global gene expression signatures using 'gene set enrichment analysis' (GSEA) (*Subramanian et al., 2005*) identified a significant enrichment of molecular signatures related to epithelial-to-mesenchymal transition, adherens, and tight junctions in the EGFP^+ fraction, characteristic of the SOX2^+ population (*Figure 3—figure supplement 1B*). The SOX2^+ fraction also displayed enrichment for genes associated with several signalling pathways known to be active in these cells, including epidermal growth factor receptor (EGFR) (*Iwai-Liao et al., 2000*), Hippo (*Lodge et al., 2016*; *Lodge et al., 2019*; *Xekouki et al., 2019*), MAPK (*Haston et al., 2017*), FGF (*Higuchi et al., 2017*), Ephrin (*Yoshida et al., 2015*; *Yoshida et al., 2017*), and p53 (*Gonzalez-Meljem et al., 2017*; *Figure 3—figure supplement 1C*, *Supplementary file 1*). Additionally, PI3K, TGFβ, and BMP pathway genes were significantly enriched in the SOX2^+ population (*Figure 3—figure supplement 1C*, *Supplementary file 1*). Query of the WNT-associated genes did not suggest a global enrichment in WNT targets (e.g. enrichment of *Myc* and *Jun*, but not of *Axin2* or *Lef1*) (*Figure 3—figure supplement 1D*, *Supplementary file 1*). Instead, SOX2^+ PSCs expressed a unique transcriptomic fingerprint of key pathway genes including *Lgr4*, *Znrf3*, *Rnf43* capable of regulating WNT signal intensity in SOX2^+ PSCs, as well as enriched expression of the receptors *Fzd1*, *Fzd3*, *Fzd4*, *Fzd6*, and *Fzd7* (*Figure 3—figure supplement 1D*). The predominant R-spondin gene expressed in the pituitary was *Rspo4*, specifically by the EGFP-negative fraction (*Figure 3—figure supplement 1D*). The gene profiling revealed that *Wls* expression encoding Gpr177/WLS, a necessary mediator of WNT ligand secretion (*Carpenter et al., 2010*; *Takeo et al., 2013*; *Wang et al., 2015*), is enriched in SOX2^+ PSCs (*Figure 3C*). Analysis of *Wnt* gene expression confirmed enriched expression of *Wnt2*, *Wnt5a,* and *Wnt9a* in SOX2^+ PSCs, and the expression of multiple additional *Wnt* genes by both fractions at lower levels (SOX2^+ fraction: *Wnt5b*, *Wnt6*, *Wnt16*; SOX2^− fraction: *Wnt2*, *Wnt2b*, *Wnt3*, *Wnt4*, *Wnt5a*, *Wnt5b*, *Wnt9a*, *Wnt10a*, *Wnt16*) (*Figure 3D*). These results reveal that SOX2^+ PSCs express the essential components to regulate activation of the WNT pathway and express *Wnt* genes as well as the necessary molecular machinery to secrete WNT ligands.

## Paracrine signalling from SOX2^+ stem cells promotes WNT activation

We sought to conclusively determine if WNT secretion specifically from SOX2^+ PSCs drives proliferation of surrounding cells in the postnatal pituitary gland. We proceeded to delete *Wls* only in the *Sox2*-expressing population (*Sox2^{CreERT2/+};Wls^{fl/fl}*) from P14 by a series of tamoxifen injections. Due to morbidity, we limited analyses to 1 week following induction. Pituitaries appeared mildly hypoplastic at P21 along the medio-lateral axis (*Figure 4—figure supplement 1*, *n* = 4 controls and *n* = 5 mutants). To determine if this is a result of reduced proliferation, we carried out immunofluorescence using antibodies against Ki-67 and SOX2. This revealed significantly fewer cycling cells in the SOX2^− population of *Sox2^{CreERT2/+};Wls^{fl/fl}* mutant pituitaries compared to *Sox2^{+/+};Wls^{fl/fl}* controls (10.326% Ki-67 in control [*n* = 4] compared to 3.129% in mutant [*n* = 5], p=0.0008, unpaired *t*-test)

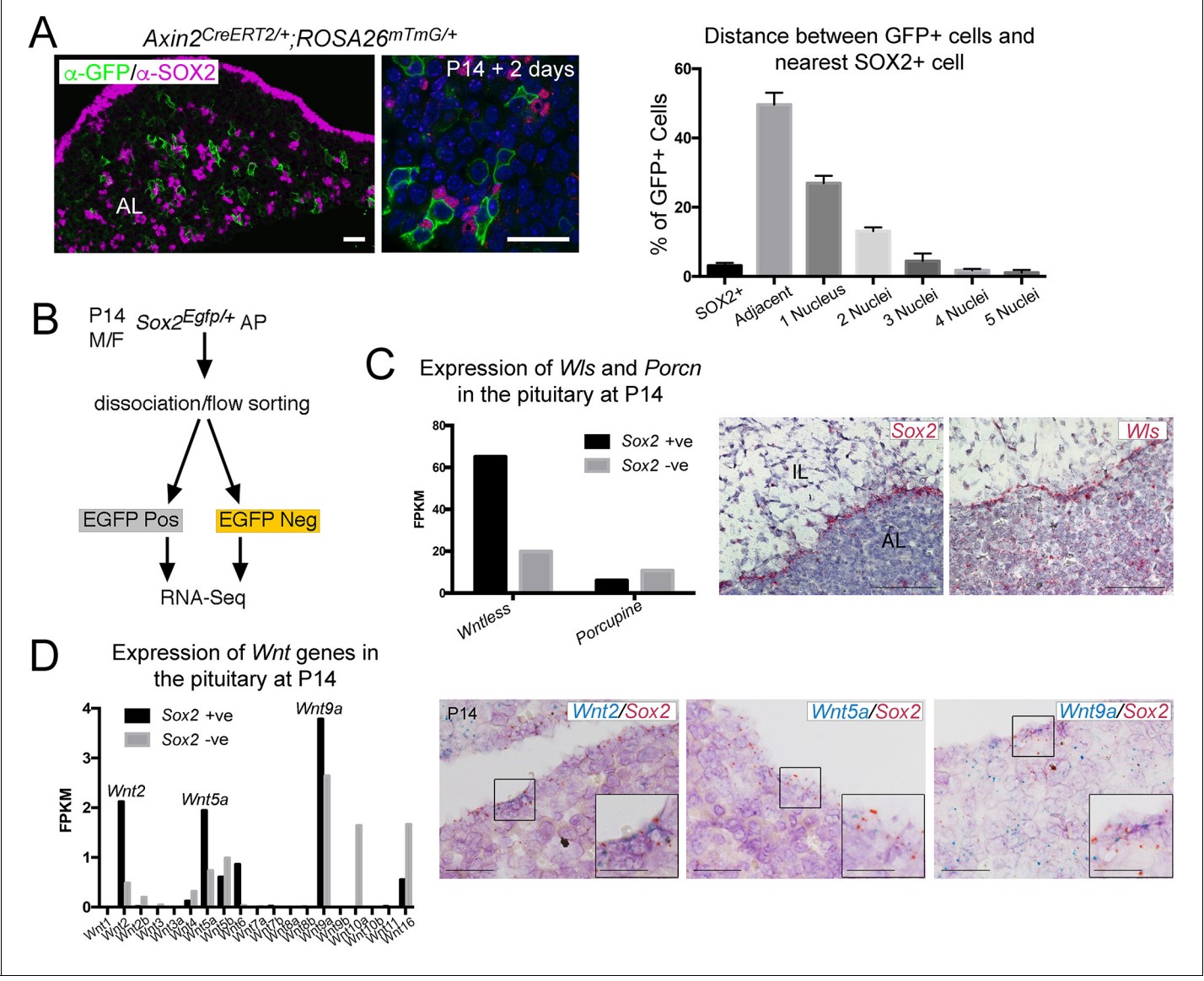

**Figure 3.** SOX2[+] pituitary stem cells (PSCs) are as a source of WNT ligands in the pituitary. (**A**) Immunofluorescence staining against GFP (green) and SOX2 (magenta) in *Axin2*[CreERT2/+]; *ROSA26*[mTmG/+] pituitaries 48 hr post-induction. Graph representing a quantification of the proximity of individual GFP[+] cells to the nearest SOX2[+] cell as quantified by the number of nuclei separating them. Plotted data represents the proportion of GFP+ cells that fall into each category of the total GFP+ cells, taken from *n* = 3 separate pituitaries. Scale bars: 50 µm. (**B**) Experimental paradigm for RNA Seq analysis of *Sox2* positive and negative cells. (**C**) Graphs representing the FPKM values of *Wls* and *Porcupine* in *Sox2* positive and negative cells (black and grey bars, respectively). mRNA in situ hybridisation for *Sox2* and for *Wls* on wild-type sagittal pituitaries at P14, demonstrating strong *Wls* expression in the marginal zone epithelium. Scale bars: 250 µm. (**D**) Bar chart showing the FPKM values of *Wnt* genes in the *Sox2*[+] and *Sox2*[−] fractions. Double mRNA in situ hybridisation against *Wnt2*, *Wnt5a*, and *Wnt9a* (blue) together with *Sox2* (red) validating expression in the *Sox2*[+] population. Boxed regions through the marginal zone epithelium are magnified. Scale bars: 100 µm and 50 µm in boxed inserts.

The online version of this article includes the following figure supplement(s) for figure 3:

**Figure supplement 1.** SOX2[+] pituitary stem cells (PSCs) are as a source of WNT ligands in the pituitary.

(*Figure 4A*). Additionally, we observed a reduction of cycling cells within the SOX2[+] population (5.582% Ki-67 in control compared to 2.225% in induced *Sox2*[CreERT2/+];*Wls*[fl/fl] mutant pituitaries, p=0.0121, unpaired *t*-test) (*Figure 4A*), resulting in a smaller SOX2[+] cell pool in mutants (23.425% SOX2[+]/total AP cells in *Sox2*[+/+];*Wls*[fl/fl] controls compared to 19.166% SOX2[+]/total AP cells in induced *Sox2*[CreERT2/+];*Wls*[fl/fl] mutant pituitaries, p=0.0238, Student's *t*-test, *n* = 5 mutants, four

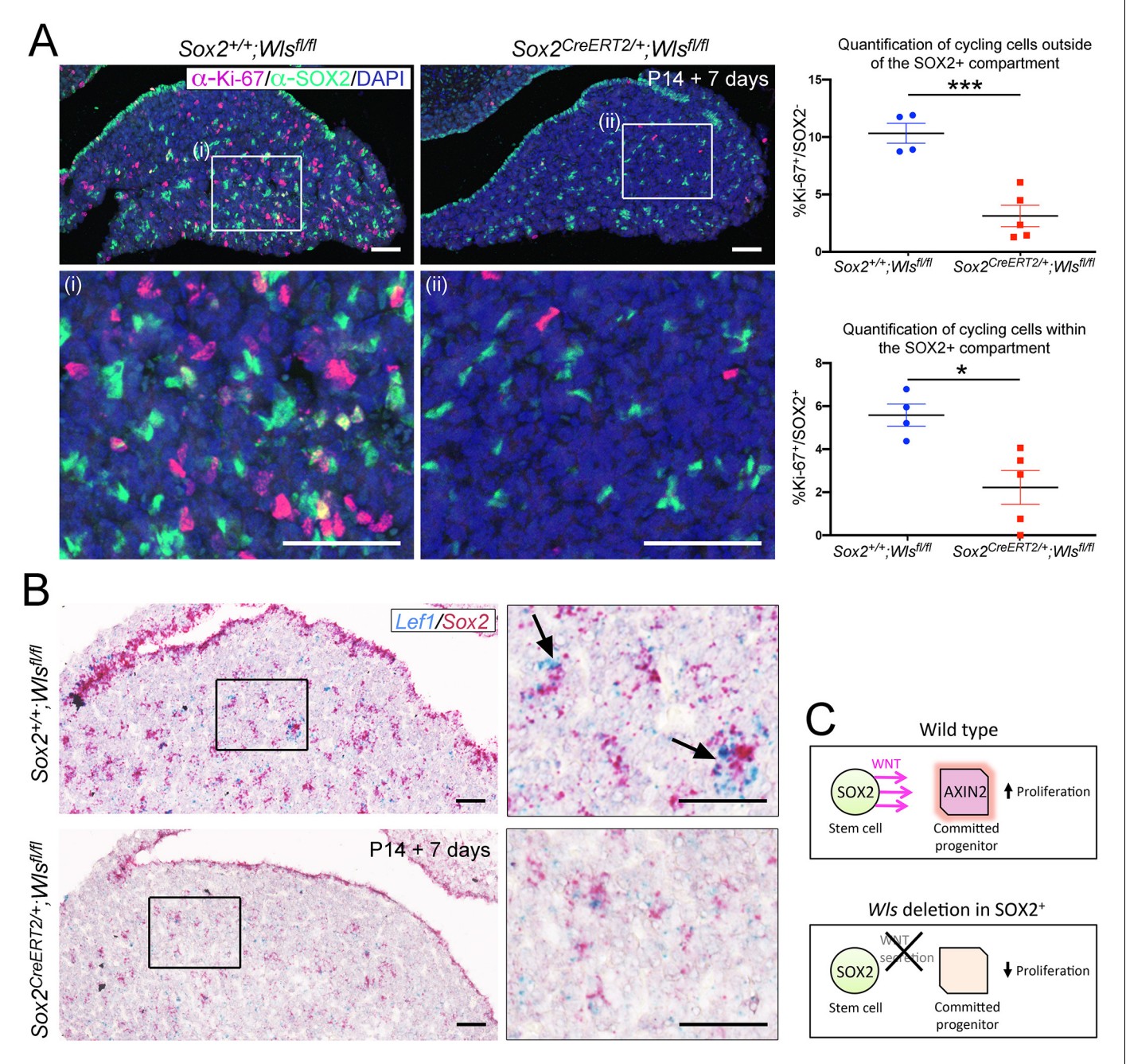

**Figure 4.** Paracrine secretion of WNTs from SOX2$^+$ pituitary stem cells (PSCs) is necessary for expansion of committed cells. (**A**) Immunofluorescence staining against SOX2 (green) and Ki-67 (magenta) in *Sox2$^{+/+}$;Wls$^{fl/fl}$* (control) and *Sox2$^{CreERT2/+}$;Wls$^{fl/fl}$* (mutant) pituitaries induced from P14 and analysed after 1 week. Nuclei were counterstained with Hoechst. (i and ii) represent magnified fields of view of regions indicated by white boxes in top panels. Scale bars: 50 µm. Graph of quantification of cycling cells marked by Ki-67 among cells negative for SOX2. Values represent mean ± SEM, p=0.0008, unpaired *t*-test. Graph of quantification of cycling cells marked by Ki-67 among SOX2-positive cells. Values represent mean ± SEM, p=0.0121, unpaired *t*-test. Each data point shows the mean of one biological replicate, *n* = 4 pituitaries from controls and five pituitaries from mutants. (**B**) Double mRNA in situ hybridisation using specific probes against *Lef1* (blue) and *Sox2* (red) in control and mutant pituitaries following tamoxifen induction from P14 and tracing for 7 days. Scale bars: 250 µm and 50 µm in boxed regions. (**C**) Model summarising paracrine WNT secretion from SOX2$^+$ PSCs to lineage-committed progenitors and the effects of abolishing WNT secretion from SOX2$^+$ PSCs through the deletion of *Wls*.

The online version of this article includes the following figure supplement(s) for figure 4:

**Figure supplement 1.** Paracrine secretion of WNTs from SOX2$^+$ pituitary stem cells (PSCs) is necessary for expansion of committed cells.

controls). To determine if reduced levels of WNT activation accompanied this phenotype, we carried out double mRNA in situ hybridisation using specific probes against *Lef1* and *Sox2*. There was an overall reduction in *Lef1* expression in mutants compared to controls ($n = 4$ per genotype), in which we frequently observed robust expression of *Lef1* transcripts in close proximity to cells expressing *Sox2* (arrows, *Figure 4B*). Together, our data support a paracrine role for SOX2$^+$ PSCs in driving the expansion of committed progeny through the secretion of WNT ligands (*Figure 4C*).

## Discussion

Emerging disparities between the archetypal stem cell model, exhibited by the haematopoietic system, and somatic stem cells of many organs have led to the concept that stem cell function can be executed by multiple cells not fitting a typical stem cell paradigm (*Clevers and Watt, 2018*). In organs with persistent populations possessing typical functional stem cell properties yet contributing minimally to turnover and repair, the necessity for such classical stem cells is questioned. Here we show that WNT signalling is required for postnatal pituitary growth by both SOX2$^+$ PSCs and SOX2$^-$ committed progenitors. We identify an additional discreet function for SOX2$^+$ PSCs, where these signal in a feedforward manner by secreting WNT ligands as cues to stimulate proliferation and promote tissue growth.

Consistent with previous reports, our data support that SOX2$^+$ PSCs contribute, but do not carry out the majority of tissue expansion during the postnatal period (*Zhu et al., 2015*); instead, new cells primarily derive from more committed progenitors, which we show to be WNT-responsive. We demonstrate that this population of lineage-restricted WNT-responsive cells rapidly expands and contributes long-lasting clones from postnatal stages. It remains to be shown if cells among the SOX2$^-$ lineage-committed populations may also fall under the classical definition of a stem cell. Preventing secretion of WNT ligands from SOX2$^+$ PSCs reveals that far from being dispensable, paracrine actions of the SOX2$^+$ population that are inactive in their majority are necessary for anterior lobe expansion from lineage-committed populations. In the adrenal gland, R-spondins are necessary for cortical expansion and zonation, where deletion of *Rspo3*, expressed by the capsule that contains adrenocortical stem cells, results in reduced proliferation of the underlying steroidogenic cells (*Vidal et al., 2016*). Corroborating a model where committed pituitary progenitors depend on the paracrine actions of SOX2$^+$ PSCs, Zhu and colleagues observed that in pituitaries with reduced numbers of PSCs, proliferation among PIT1$^+$ cells was significantly impaired (*Zhu et al., 2015*). It would be intriguing to see if there is a reduction in WNT signalling in this model, or following genetic ablation of adult SOX2$^+$ PSCs (*Roose et al., 2017*).

We show that a subpopulation of SOX2$^+$ PSCs in the postnatal gland are also WNT-responsive and have greater in vitro colony-forming potential under defined conditions. This colony-forming potential is normally a property of a minority of SOX2$^+$ PSCs at any given age and reflects their in vivo proliferative capacity (*Andoniadou et al., 2012*; *Rizzoti et al., 2013*). A role for the WNT pathway in promoting SOX2$^+$ cell activity is supported by studies showing that pathogenic overexpression of β-catenin promotes their colony-forming ability (*Sarkar et al., 2016*) and their in vivo expansion (*Andoniadou et al., 2012*). Additionally, elevated WNT pathway activation has been described for pituitary side-population cells, enriched for SOX2$^+$ stem cells from young, compared to old pituitaries (*Gremeaux et al., 2012*). This is in line with our findings that the WNT pathway has an important function in promoting the activation of SOX2$^+$ PSCs. It remains to be shown if this response relies on autocrine WNT-signalling as for other stem cells (*Lim et al., 2013*); however, our results reveal reduced proliferation among SOX2$^+$ PSCs and reduced SOX2$^+$ cell numbers when WNT secretion from these cells is abolished, supportive of either autocrine signalling or paracrine signalling between different subsets of the SOX2$^+$ population.

The mechanism preventing the majority of SOX2$^+$ PSCs from responding to WNT signals remains elusive but points to heterogeneity among the population. Such regulation could occur at the level of receptor signalling; we have shown by bulk transcriptomic profiling that SOX2$^+$ PSCs express the receptors required to respond to the WNT pathway, but also express high levels of the frizzled inhibitor *Znrf3*, and the R-spondin receptor *Lgr4*. One conceivable scenario is that high levels of *Znrf3* inhibit frizzled receptors in the absence of R-spondin under normal physiological conditions, supressing a WNT response. In support of this, R-spondins have been shown to promote pituitary organoid formation (*Cox et al., 2019*). Whether the R-spondin/LGR/ZNRF3 module is active under

physiological conditions needs to be determined. Furthermore, well-described factors expressed in PSCs are known to have inhibitory effects on β-catenin-mediated transcription, such as YAP/TAZ (*Azzolin et al., 2014*; *Gregorieff et al., 2015*) and SOX2 itself (*Alatzoglou et al., 2011*; *Kelberman et al., 2008*).

In summary, we demonstrate an alternative mechanism for stem cell contribution to homeostasis, whereby these can act as paracrine signalling hubs to promote local proliferation. Applicable to other organs, this missing link between SOX2[+] PSCs and committed cell populations of the AP is key for basic physiological functions and renders stem cells integral to organ expansion.

# Materials and methods

**Key resources table**

| Reagent type (species) or resource | Designation | Source or reference | Identifiers | Additional information |
|---|---|---|---|---|
| Genetic reagent (*Mus musculus*) | *Axin2$^{CreERT2/+}$* | Roel Nusse, Stanford University The Jackson Laboratory | JAX:018867, RRID:IMSR_JAX:018867 | |
| Genetic reagent (*Mus musculus*) | *Sox2$^{CreERT2/+}$* | (*Andoniadou et al., 2013*) PMID:24094324 DOI: 10.1016/j.stem. 2013.07.004 | MGI:5512893 | |
| Genetic reagent (*Mus musculus*) | *ROSA 26$^{mTmG/mTmG}$* | The Jackson Laboratory | JAX:007676, RRID:IMSR_JAX:007676 | |
| Genetic reagent (*Mus musculus*) | *ROSA 26$^{Confetti/Confetti}$* | The Jackson Laboratory | JAX:017492, RRID:IMSR_JAX:017492 | |
| Genetic reagent (*Mus musculus*) | *ROSA 26$^{tdTomato/tdTomato}$* | The Jackson Laboratory | JAX:007909, RRID:IMSR_JAX:007909 | |
| Genetic reagent (*Mus musculus*) | *Ctnnb1$^{fl(ex2-6)/}$ $^{fl(ex2-6)}$* (*Ctnnb$^{LOF/LOF}$*) | The Jackson Laboratory | JAX:004152, RRID:IMSR_JAX:004152 | |
| Genetic reagent (*Mus musculus*) | *Wls$^{fl/fl}$* | The Jackson Laboratory | JAX:012888, RRID:IMSR_JAX:012888 | |
| Genetic reagent (*Mus musculus*) | *Sox2$^{eGFP/+}$* | *Ellis et al., 2004* PMID:15711057 DOI: 10.1159/000082134 | MGI:3589809 | |
| Genetic reagent (*Mus musculus*) | TCF/Lef:H2B-GFP | The Jackson Laboratory | JAX:013752, RRID:IMSR_JAX:013752 | |
| Cell line (*Mus musculus*) | Primary anterior pituitary cells | This paper | N/A | Freshly isolated from *Mus musculus.* |
| Antibody | Anti-GFP (Chicken Polyclonal) | Abcam | ab13970, RRID:AB_300798 | IF(1:400) |
| Antibody | Anti-SOX2 (Goat Polyclonal) | Immune Systems Ltd | GT15098, RRID:AB_2195800 | IF(1:200) |
| Antibody | Anti-SOX2 (Rabbit Monoclonal) | Abcam | ab92494, RRID:AB_10585428 | IF(1:100) |
| Antibody | Anti-SOX9 (Rabbit Monoclonal) | Abcam | ab185230, RRID:AB_2715497 | IF(1:500) |
| Antibody | Anti-POU1F1 (PIT1) (Rabbit Monoclonal) | Gifted by Dr S. J. Rhodes (IUPUI, USA) | 422_Rhodes, RRID:AB_2722652 | IF(1:500) |
| Antibody | Anti-SF1 (NR5A1, clone N1665) (Mouse Monoclonal) | Thermo Fisher Scientific | 434200, RRID:AB_2532209 | IF(1:300) |

*Continued on next page*

*Continued*

| Reagent type (species) or resource | Designation | Source or reference | Identifiers | Additional information |
|---|---|---|---|---|
| Antibody | Anti-TBX19 (TPIT), (Rabbit Polyclonal) | Gifted by Dr J. Drouin (Montreal Clinical Research Institute, Canada) | Ac1250 #71, RRID:AB_2728662 | IF(1:200) |
| Antibody | Anti-Ki67 (Rabbit Monoclonal) | Abcam | ab15580, RRID:AB_443209 | IF(1:100) |
| Antibody | Anti-pH-H3 (Rabbit Polyclonal) | Abcam | ab5176, RRID:AB_304763 | IF(1:500) |
| Antibody | Anti-GH (Rabbit Polyclonal) | National Hormone and Peptide Program (NHPP) | AFP-5641801 | IF(1:1000) |
| Antibody | Anti-TSH (Rabbit Polyclonal) | National Hormone and Peptide Program (NHPP) | AFP-1274789 | IF(1:1000) |
| Antibody | Anti-PRL (Rabbit Polyclonal) | National Hormone and Peptide Program (NHPP) | AFP-4251091 | IF(1:1000) |
| Antibody | Anti-ACTH (Mouse Monoclonal) | Fitzgerald | 10C-CR1096M1, RRID:AB_1282437 | IF(1:400) |
| Antibody | Anti-LH (Rabbit Polyclonal) | National Hormone and Peptide Program (NHPP) | AFP-697071P | IF(1:300) |
| Antibody | Anti-FSH (Rabbit Polyclonal) | National Hormone and Peptide Program (NHPP) | AFP-HFS6 | IF(1:300) |
| Antibody | Anti-ZO-1 (Rat Monoclonal) | Santa Cruz | SC33725, RRID:AB_628459 | IF(1:300) |
| Antibody | Anti-E-CADHERIN (Rabbit Monoclonal) | Cell Signaling | 3195S, RRID:AB_2291471 | IF(1:300) |
| Antibody | Anti-Rabbit 488 (Goat Polyclonal) | Life Technologies | A11008, RRID:AB_143165 | IF(1:400) |
| Antibody | Anti-Rabbit 555 (Goat Polyclonal) | Life Technologies | A21426, RRID:AB_1500929 | IF(1:400) |
| Antibody | Anti-Rabbit 633 (Goat Polyclonal) | Life Technologies | A21050, RRID:AB_141431 | IF(1:400) |
| Antibody | Anti-Goat 488 (Donkey Polyclonal) | Abcam | ab150133, RRID:AB_2832252 | IF(1:400) |
| Antibody | Anti-Chicken 488 (Goat Polyclonal) | Life Technologies | A11039, RRID:AB_142924 | IF(1:400) |
| Antibody | Anti-Chicken 647 (Goat Polyclonal) | Life Technologies | A21449, RRID:AB_1500594 | IF(1:400) |
| Antibody | Anti-Rat 555 (Goat Polyclonal) | Life Technologies | A21434, RRID:AB_141733 | IF(1:400) |
| Antibody | Anti-Mouse 555 (Goat Polyclonal) | Life Technologies | A21426, RRID:AB_1500929 | IF(1:400) |
| Antibody | Anti-Rabbit Biotinylated (Donkey Polyclonal) | Abcam | ab6801, RRID:AB_954900 | IF(1:400) |
| Antibody | Anti-Rabbit Biotinylated (Goat Polyclonal) | Abcam | ab207995 | IF(1:400) |

*Continued on next page*

*Continued*

| Reagent type (species) or resource | Designation | Source or reference | Identifiers | Additional information |
|---|---|---|---|---|
| Antibody | Anti-Mouse Biotinylated (Goat Biotinylated) | Abcam | ab6788, RRID:AB_954885 | IF(1:400) |
| Sequence-based reagent | RNAscope probe *M. musculus Axin2* | Advanced Cell Diagnostics | 400331 | |
| Sequence-based reagent | RNAscope probe *M. musculus Lef1* | Advanced Cell Diagnostics | 441861 | |
| Sequence-based reagent | RNAscope probe *M. musculus Wls* | Advanced Cell Diagnostics | 405011 | |
| Sequence-based reagent | RNAscope probe *M. musculus Rspo1* | Advanced Cell Diagnostics | 401991 | |
| Sequence-based reagent | RNAscope probe *M. musculus Rspo2* | Advanced Cell Diagnostics | 402001 | |
| Sequence-based reagent | RNAscope probe *M. musculus Rspo3* | Advanced Cell Diagnostics | 402011 | |
| Sequence-based reagent | RNAscope probe *M. musculus Rspo4* | Advanced Cell Diagnostics | 402021 | |
| Sequence-based reagent | RNAscope probe *M. musculus Lgr4* | Advanced Cell Diagnostics | 318321 | |
| Sequence-based reagent | RNAscope probe *M. musculus Wnt9a* | Advanced Cell Diagnostics | 405081 | |
| Sequence-based reagent | RNAscope probe *M. musculus Wnt2* | Advanced Cell Diagnostics | 313601 | |
| Sequence-based reagent | RNAscope probe *M. musculus Wnt5a* | Advanced Cell Diagnostics | 316791 | |
| Sequence-based reagent | RNAscope probe *eGFP* | Advanced Cell Diagnostics | 400281 | |
| Sequence-based reagent | RNAscope probe *M. musculus Jun* | Advanced Cell Diagnostics | 453561 | |
| Sequence-based reagent | RNAscope probe *M. musculus Axin2* (Channel 2) | Advanced Cell Diagnostics | 400331-C2 | |
| Sequence-based reagent | RNAscope probe *M. musculus Sox2* (Channel 2) | Advanced Cell Diagnostics | 401041-C2 | |
| Sequence-based reagent | RNAscope probe *eGFP* (Channel 2) | Advanced Cell Diagnostics | 400281-C2 | |
| Sequence-based reagent | RNAscope probe *M. musculus Sox2* | Advanced Cell Diagnostics | 401041 | |
| Sequence-based reagent | RNAscope probe *M. musculus Pou1f1* | Advanced Cell Diagnostics | 486441 | |
| Sequence-based reagent | RNAscope probe Duplex Positive Control *Ppib-C1, Polr2a-C2* | Advanced Cell Diagnostics | 321641 | |
| Sequence-based reagent | RNAscope probe Duplex Negative Control *DapB* (both channels) | Advanced Cell Diagnostics | 320751 | |
| Sequence-based reagent | RNAscope probe Singleplex Positive Control *Ppib* | Advanced Cell Diagnostics | 313911 | |

*Continued on next page*

*Continued*

| Reagent type (species) or resource | Designation | Source or reference | Identifiers | Additional information |
|---|---|---|---|---|
| Sequence-based reagent | RNAscope probe: Singleplex Negative Control *DapB* | Advanced Cell Diagnostics | 310043 | |
| Peptide, recombinant protein | Streptavidin 488 | Life Technologies | S11223 | IF(1:400) |
| Peptide, recombinant protein | Streptavidin 555 | Life Technologies | S32355 | IF(1:400) |
| Peptide, recombinant protein | Streptavidin 633 | Life Technologies | S21375 | IF(1:400) |
| Commercial assay or kit | RNAScope 2.5 HD Assay-RED | Advanced Cell Diagnostics | 322350 | |
| Commercial assay or kit | RNAScope 2.5 HD Duplex Assay | Advanced Cell Diagnostics | 322430 | |
| Commercial assay or kit | LIVE/DEAD Fixable Near IR-Dead Cell Stain Kit | Life Technologies | L34975 | |
| Commercial assay or kit | FIX and PERM Cell Permeabilization Kit | Life Technologies | GAS003 | |
| Chemical compound, drug | Tamoxifen | Sigma | T5648 | |
| Chemical compound, drug | Corn Oil | Sigma | C8267 | |
| Chemical compound, drug | Collagenase Type 2 | Worthington | 4178 | C |
| Chemical compound, drug | 10× Trypsin | Sigma | 59418C | |
| Chemical compound, drug | Deoxyribonuclease I | Worthington | LS002172 | |
| Chemical compound, drug | Fungizone | Gibco | 15290 | |
| Chemical compound, drug | Hank's Balanced Salt Solution (HBSS) | Gibco | 14170 | |
| Chemical compound, drug | Foetal Bovine Serum | Sigma | F2442 | |
| Chemical compound, drug | HEPES | Thermo Fisher | 15630 | |
| Chemical compound, drug | bFGF | R&D Systems | 233-FB-025 | |
| Chemical compound, drug | Cholera Toxin | Sigma | C8052 | |

*Continued on next page*

*Continued*

| Reagent type (species) or resource | Designation | Source or reference | Identifiers | Additional information |
|---|---|---|---|---|
| Chemical compound, drug | DMEM-F12 | Thermo Fisher | 31330 | |
| Chemical compound, drug | Penicillin/ Streptomycin | Gibco | 15070-063 | |
| Chemical compound, drug | Neutral Buffered Formalin | Sigma | HT501128 | |
| Chemical compound, drug | Hoechst 33342 | Thermo Fisher | H3570 | 1:1000 |
| Chemical compound, drug | Declere | Sigma | D3565 | |
| Chemical compound, drug | Neo-Clear | Sigma | 65351-M | |
| Software, algorithm | FlowJo | FlowJo, LLC | https://www.flowjo.com/ RRID:SCR_008520 | |
| Software, algorithm | Prism 7 | GraphPad Software | https://www.graphpad.com/ | |
| Software, algorithm | Image Lab | Bio-Rad Laboratories | http://www.bio-rad.com/ | |
| Software, algorithm | NDP View | Hamamatsu Photonics | https://www.hamamatsu.com/ | |
| Software, algorithm | HISAT v2.0.3 | *Kim et al., 2015* | https://github.com/infphilo/hisat2 RRID:SCR_015530 | |
| Software, algorithm | DESeq2 v2.11.38 | *Love et al., 2014* | https://github.com/Bioconductor-mirror/DESeq2 RRID:SCR_015687 | |
| Software, algorithm | featureCounts v1.4.6p5 | *Liao et al., 2014* | http://subread.sourceforge.net/ RRID:SCR_012919 | |
| Software, algorithm | The Galaxy Platform | *Afgan et al., 2016*; *Blankenberg et al., 2010*; *Goecks et al., 2010* | https://usegalaxu.org RRID:SCR_006281 | |
| Software, algorithm | Gene Set Enrichment Analysis (GSEA) | *Subramanian et al., 2005* | software.broadinstitute.org/gsea/index.jsp RRID:SCR_003199 | |
| Software, algorithm | Cufflinks | *Trapnell et al., 2012* | https://github.com/cole-trapnell-lab/cufflinks RRID:SCR_014597 | |
| Other | Deposited Data, RNA-Seq | BioProject (NCBI) | PRJNA421806 | |

## Mice

All procedures were performed under compliance of the Animals (Scientific Procedures) Act 1986, Home Office License (P5F0A1579). KCL Biological Services Unit staff undertook daily animal husbandry. Genotyping was performed on ear biopsies taken between P11 and P15 by standard PCR using the indicated primers. These experiments were not conducted at random and the experimenters were not blind while conducting the animal handling and assessment of tissue. Images are

representative of the respective genotypes. For all studies, both male and female animals were used and results combined.

The $Sox2^{CreERT2/+}$ and $Sox2^{Egfp/+}$ strains were kept on a CD-1 background. $Axin2^{CreERT2/+}$ animals were kept on a mixed background of C57BL/6 backcrossed onto CD-1 for five generations and were viable and fertile, with offspring obtained at the expected Mendelian ratios. $ROSA26^{mTmG/mTmG}$, $ROSA26^{Confetti/Confetti}$, $ROSA26^{tdTomato/tdTomato}$, $Wls^{fl/fl}$, $Ctnnb1^{fl(ex2-6)/\ fl(ex2-6)}$, and TCF/LEF:H2B-EGFP mice were kept on a mixed background of 129/Sv backcrossed onto CD-1 for at least three generations. For lineage tracing studies, male $Axin2^{CreERT2/+}$ or $Sox2^{CreERT2/+}$ mice were bred with homozygous $ROSA26^{mTmG/mTmG}$ or $ROSA26^{Confetti/Confetti}$ dams to produce the appropriate allele combinations on the reporter background. Pups were induced at P14 or P15 with a single dose of tamoxifen (resuspended to 20 mg/ml in Corn Oil with 10% ethanol) by intraperitoneal injection, at a concentration of 0.15 mg/g of body weight. Pituitaries were harvested at the indicated time points post-induction and processed for further analysis as described below. Mice were harvested from different litters for each time point at random. For litters in which there was a surplus of experimental mice, multiple samples were harvested for each required time point.

For Wntless deletion studies, $Sox2^{CreERT2/+};Wls^{fl/+};ROSA26^{mTmG/mTmG}$ males were bred with $Wls^{fl/fl};ROSA26^{mTmG/mTmG}$ dams, to produce $Sox2^{CreERT2/+};Wls^{fl/+};ROSA26^{mTmG/mTmG}$, $Sox2^{CreERT2/+};Wls^{fl/fl};ROSA26^{mTmG/mTmG}$, and $Wls^{fl/fl};ROSA26^{mTmG/mTmG}$ offspring. Pups of the indicated genotypes received intraperitoneal injections of 0.15 mg of tamoxifen per gram body weight on four consecutive days, beginning at P14, and harvested 3 days after the final injection.

For the β-catenin loss-of-function experiments, either $Sox2^{CreERT2/+};Ctnnb1^{fl(ex2-6)/+};ROSA26^{mTmG/mTmG}$ or $Axin2^{CreERT2/+};Ctnnb1^{fl(ex2-6)/+};ROSA26^{mTmG/mTmG}$ males were crossed with $Ctnnb1^{fl(ex2-6)/fl(ex2-6)};ROSA26^{mTmG/mTmG}$ dams. $Axin2^{CreERT2/+};Ctnnb1^{fl(ex2-6)/fl(ex2-6)};ROSA26^{mTmG/mTmG}$ and $Axin2^{CreERT2/+};Ctnnb1^{fl(ex2-6)/+};ROSA26^{mTmG/mTmG}$ pups were induced with a single dose of tamoxifen, at a concentration of 0.15 mg/g of body weight and kept alive for 7 days before harvesting. $Sox2^{CreERT2/+};Ctnnb1^{fl(ex2-6)/+};ROSA26^{mTmG/mTmG}$ and $Sox2^{CreERT2/+};Ctnnb1^{fl(ex2-6)/fl(ex2-6)};ROSA26^{mTmG/mTmG}$ pups received two intraperitoneal injections of tamoxifen, at a concentration of 0.15 mg/g of body weight, on two consecutive days and were kept alive for the indicated length of time before harvesting.

TCF/LEF:H2B-EGFP mice were culled and the pituitaries harvested at the indicated ages for the respective experiments. For FACS experiments, mice were harvested at 21 days of age. $Axin2^{CreERT2/+};Sox2^{eGFP/+}$ males were crossed with $ROSA26^{tdTomat/tdTomato}$ dams to produce $Axin2^{CreERT2/+};Sox2^{eGFP/+};ROSA26^{tdTomato/+}$ that were induced with single doses of tamoxifen at 21 and 22 days of age and harvested 3 days after the first injection for FACS experiments.

## Flow cytometry analysis of lineage traced pituitaries

For the quantification of cells by flow cytometry, anterior lobes of $Axin2^{CreERT2/+};ROSA26^{mTmG/+}$ mice dissected at the indicated time points. The posterior and intermediate lobes were dissected from the anterior lobes under a dissection microscope. Untreated $ROSA26^{mTmG/+}$ and wild-type pituitaries from age-matched litters were used as tdTomato only and negative controls, respectively. Dissected pituitaries were incubated in Enzyme Mix (0.5% w/v collagenase type 2 [Lorne Laboratories], 0.1× Trypsin [Gibco], 50 µg/ml DNase I [Worthington], and 2.5 µg/ml Fungizone [Gibco] in Hank's Balanced Salt Solution [HBSS] [Gibco]) in a cell culture incubator for up to 3 hr; 850 ml of HBSS was added to each Eppendorf in order to quench the reaction. Pituitaries were dissociated by agitation, pipetting up and down 100× at first with a 1 ml pipette, followed by 100× with a 200 µl pipette. Cells were transferred to a 15 ml Falcon tube and resuspended in 9 ml of HBSS and spun down at 200 g for 5 min. The supernatant was aspirated, leaving behind the cell pellet that was resuspended in PBS and spun down at 1000 rpm for 5 min before being resuspended in a Live/Dead cell stain (Life Technologies, L34975) prepared to manufacturer's instructions, for 30 min in the dark. Cells were washed in PBS as above. The pellet was resuspended in FIX and PERM Cell Permeabilization Kit (Life Technologies, GAS003) prepared as per manufacturer's instructions for 10 min at room temperature. Cells were washed as above, and the pellet was resuspended in 500 µl of FACS buffer (1% foetal calf serum [Sigma], 25 mM HEPES in PBS) and filtered through 70 µm filters (BD Falcon), into 5 ml round bottom polypropylene tubes (BD Falcon). One minute prior to analysis, 1 µl of Hoechst was added to the suspended cells and incubated. Samples were analysed on a BD Fortessa and gated according to negative and single fluorophore controls. Single cells were gated according

to SSC-A and SSC-W. Dead cells were excluded according to DAPI (2 ng/ml, incubated for 2 min prior to sorting). GFP$^+$, tdTomato$^+$, and GFP$^+$;tdTomato$^+$ cells were gated according to negative controls in the PE-A and FITC-A channels.

## FACS for sequencing or colony forming assays

For FACS, the anterior lobes from *Sox2$^{eGFP/+}$*, TCF/LEF:H2B-GFP, or *Axin2$^{CreERT2/+}$;Sox2$^{eGFP/+}$; ROSA26$^{tdTomato/+}$* and their respective controls were dissected and dissociated as above. After dissociation cells were spun down at 200 g in HBSS and the pellet was resuspended in 500 µl FACS buffer. Using an Aria III FACs machine (BD systems), samples were gated according to negative controls, and where applicable single fluorophore controls. Experimental samples were sorted according to their fluorescence, as indicated, into tubes containing either RNAlater (Qiagen) for RNA isolation or 1 ml of Pit Complete Media for culture (Pit Complete: 20 ng/ml) bFGF and 50 ng/ml of cholera toxin in 'Pit Basic' media (DMEM-F12 with 5% foetal calf serum, 100 U/ml penicillin, and 100 µg/ml streptomycin). Cells were plated in 12-well plates at clonal density, approximately 500 cells/well. Colonies were incubated for a total of 7 days before being fixed in 10% neutral buffered formalin (NBF) (Sigma) for 10 min at room temperature, washed for 5 min, three times, with PBS and stained with crystal violet in order for the number of colonies to be quantified.

## RNA-sequencing

Total RNA was isolated from each sample and following poly-A selection, cDNA libraries were generated using TruSeq (Clontech, 634925). Barcoded libraries were then pooled at equal molar concentrations and sequenced on an Illumina Hiseq 4000 instrument in a 75 base pair, paired-end sequencing mode, at the Wellcome Trust Centre for Human Genetics (Oxford, United Kingdom). Raw sequencing reads were quality checked for nucleotide calling accuracy and trimmed accordingly to remove potential sequencing primer contaminants. Following QC, forward and reverse reads were mapped to GRCm38/mm10 using Hisat2 (*Kim et al., 2015*). Using a mouse transcriptome specific GTF as a guide, FeatureCounts (*Liao et al., 2014*) was used to generate gene count tables for every sample. These were utilised within the framework of the Deseq2 (*Love et al., 2014*) and FPKM values (generated by FPKM count *Wang et al., 2012*) were processed using the Cufflinks (*Trapnell et al., 2012*) pipelines that identified statistically significant gene expression differences between the sample groups. Following identification of differentially expressed genes (at an FDR < 0.05) we focused on identifying differentially expressed pathways using a significance threshold of FDR < 0.05 unless otherwise specified. The gene lists used for GSEA were as found on the BROAD institute GSEA MSigDBv.7 'molecular signatures database'. The deposited data set (BioProject, accession PRJNA421806) can be accessed through the following link: https://www.ncbi.nlm.nih.gov/bioproject/PRJNA421806.

## Immunofluorescence and microscopy

Freshly harvested pituitaries were washed in PBS for 10 min before being fixed in 10% NBF for 18 hr at room temperature. In short, embryos and whole pituitaries were washed in PBS three times, before being dehydrated through a series of 1 hr washes in 25%, 50%, 70%, 80%, 90%, 95%, and 100% ethanol. Tissues were washed in Neo-Clear (Sigma) at room temperature for 10 min, then in fresh preheated Neo-Clear at 60°C for 10 min. Subsequently, tissues were incubated in a mixture of 50% Neo-Clear:50% paraffin wax at 60°C for 15 min followed by three changes of pure wax for a minimum of 1 hr washes at 60°C, before being orientated to be sectioned in the frontal plane. Embedded samples were sectioned at 5 µm and mounted on to Super Frost+ slides.

For immunofluorescence, sections were deparaffinised in Neo-Clear by three washes of 10 min, washed in 100% ethanol for three times 5 min, and rehydrated in a series of 5-min ethanol washes up to distilled water (95%, 90%, 80%, 70%, 50%, 25%, H$_2$O). Heat induced epitope retrieval was performed with 1× DeClear Buffer (citrate pH 6) in a Decloaking chamber NXGEN (Menarini Diagnostics) for 3 min at 110°C. Slides were left to cool to room temperature before proceeding to block for 1 hr at room temperature in blocking buffer (0.2% BSA, 0.15% glycine, 0.1% TritonX in PBS) with 10% serum (sheep or donkey, depending on secondary antibodies). Primary antibodies were diluted in blocking buffer with 1% of the appropriate serum and incubated overnight at 4°C. Slides were washed three times for 10 min with PBST. Slides were incubated with secondary antibodies diluted

1:400 in blocking buffer with 1% serum for 1 hr at room temperature. Slides were washed three times with PBST as above. Where biotinylated secondary antibodies were used, slides were incubated with streptavidin diluted 1:400 in blocking buffer with 1% serum for 1 hr at room temperature. Finally, slides were washed with PBST and mounted using Vectashield Antifade Mounting Medium (Vector Laboratories, H-1000).

The following antibodies, along with their dilutions and detection technique, were used: GFP (1:400, Alexa Fluor-488 or −647 secondary), SOX2 raised in goat (1:200, Alexa Fluor-488 secondary), SOX2 raised in rabbit (1:100, biotinylated secondary), SOX9 (1:500, biotinylated secondary), PIT1 (1:500, biotinylated secondary), SF1 (1:300, biotinylated secondary), TPIT (1:200, biotinylated secondary), Ki-67 (1:100, biotinylated secondary), pH-H3 (1:500, biotinylated secondary), GH (1:1000, biotinylated secondary), TSH (1:1000, biotinylated secondary), PRL (1:1000, biotinylated secondary), ACTH (1:400, Alexa Fluor-555 secondary), LH/FSH (1:300, biotinylated secondary), ZO-1 (1:300, Alexa Fuor-488), and E-Cadherin (1:300, Alexa Fluor-488). Nuclei were visualised with Hoechst (1:1000). Images were taken on a TCS SPS Confocal (Leica Microsystem) with a 20× objective for analysis.

## mRNA in situ hybridisation

All mRNA in situ hybridisations were performed using the RNAscope singleplex or duplex chromogenic kits (Advanced Cell Diagnostics) on formalin fixed paraffin embedded sections processed as described in the above section. The protocol followed the manufacturer's instructions with slight modifications. ImmEdge Hydrophobic Barrier PAP Pen (Vector Laboratories, H-4000) was used to draw a barrier around section while air-drying following the first ethanol washes. Pretreatment followed the standard length of time for pituitaries (12 min), while embryos were boiled for 10 min. For singleplex, the protocol proceeded to follow the instructions exactly. For duplex, Amplification nine was extended to 1 hr and the dilution of the Green Detection reagent was increased to 1:30. For both protocols, sections were counterstained with Mayer's Haematoxylin (Vector Laboratories, H-3404), left to dry at 60°C for 30 min before mounting with VectaMount Permanent Mounting Medium (Vector Laboratories, H-5000). Slides were scanned using a Nanozoomer-XR Digital Slide Scanner (Hamamatsu) and processed using Nanozoomer Digital Pathology View (Hamamatsu).

## Quantification of cells

Cell numbers were quantified in ImageJ using the cell counter plugin (*Schindelin et al., 2012*). At a minimum, three sections per pituitary were quantified, spaced no less than 100 μM apart in the tissue.

## Statistics

All statistical analyses were performed in GraphPad Prism. Data points in graphs represent the mean values of recordings from a single biological replicate unless otherwise stated.

## Acknowledgements

This study has been supported by the Medical Research Council (MR/L016729/1, MR/T012153/1) (CLA), The Lister Institute of Preventive Medicine (CLA), the Deutsche Forschungsgemeinschaft (DFG German Research Foundation) (Project Number 314061271 – TRR 205) (CLA), the Howard Hughes Medical Institute (RN), the Agence Nationale de la Recherche (ANR-18-CE14-0017), and Fondation pour la Recherche Médicale (DEQ20150331732) (PM). JPR was supported by a Dianna Trebble Endowment Fund Dental Institute Studentship, EJL by the King's Bioscience Institute and the Guy's and St Thomas' Charity Prize PhD Programme in Biomedical and Translational Science, and YK by a Project Support Grant from the British Society for Neuroendocrinology. We thank Dr AF Parlow and the National Hormone and Peptide Program (Harbor–University of California, Los Angeles Medical Center) for providing some of the antibodies used in this study and Prof. J Drouin and Prof. S Rhodes for TPIT and PIT1 antibodies respectively. We thank the High-Throughput Genomics Group at the Wellcome Trust Centre for Human Genetics (funded by Wellcome Trust grant reference 090532/Z/09/Z) for the generation of the sequencing data. For flow sorting and analysis, this research was supported by the National Institute for Health Research (NIHR) Biomedical Research Centre based at Guy's and St Thomas' NHS Foundation Trust and King's College London. We thank

Marie Isabelle Garcia, Juan Pedro Martinez-Barbera, and Paul Le Tissier for useful discussions and critical comments on the manuscript.

## Additional information

### Competing interests

Roel Nusse: Reviewing editor, *eLife*. The other authors declare that no competing interests exist.

### Funding

| Funder | Grant reference number | Author |
|---|---|---|
| Medical Research Council | MR/L016729/1 | Cynthia Lilian Andoniadou |
| Medical Research Council | MR/T012153/1 | Cynthia Lilian Andoniadou |
| Deutsche Forschungsgemeinschaft | 314061271 - TRR 205 | Cynthia Lilian Andoniadou |
| Howard Hughes Medical Institute | | Roel Nusse |
| Agence Nationale de la Recherche | ANR-18-CE14-0017 | Patrice Mollard |
| Fondation pour la Recherche Médicale | DEQ20150331732 | Patrice Mollard |
| Lister Institute of Preventive Medicine | | Cynthia Lilian Andoniadou |
| Dianna Trebble Endowment Fund | | John P Russell |
| King's Bioscience Institute and the Guy's and St Thomas' Charity Prize | | Emily J Lodge |
| British Society for Neuroendocrinology | | Yasmine Kemkem |

The funders had no role in study design, data collection and interpretation, or the decision to submit the work for publication.

### Author contributions

John P Russell, Conceptualization, Formal analysis, Investigation, Methodology, Writing - original draft, Writing - review and editing; Xinhong Lim, Resources, Writing - review and editing; Alice Santambrogio, Formal analysis, Investigation; Val Yianni, Software, Formal analysis, Investigation; Yasmine Kemkem, Matthew Fish, Resources, Investigation; Bruce Wang, Anaëlle Grabek, Resources; Scott Haston, Shirleen Hallang, Emily J Lodge, Amanda L Patist, Investigation; Andreas Schedl, Resources, Supervision; Patrice Mollard, Roel Nusse, Resources, Supervision, Funding acquisition, Methodology, Writing - review and editing; Cynthia L Andoniadou, Conceptualization, Supervision, Funding acquisition, Investigation, Methodology, Writing - original draft, Writing - review and editing

### Author ORCIDs

Xinhong Lim http://orcid.org/0000-0002-4725-5161
Val Yianni http://orcid.org/0000-0001-9857-7577
Scott Haston http://orcid.org/0000-0003-3928-4808
Emily J Lodge http://orcid.org/0000-0003-0932-8515
Patrice Mollard http://orcid.org/0000-0002-2324-7589
Cynthia L Andoniadou https://orcid.org/0000-0003-4311-5855

## Ethics

Animal experimentation: This study was performed under compliance of the Animals (Scientific Procedures) Act 1986, Home Office License (P5F0A1579) and KCL Biological Safety approval for project 'Function and Regulation of Pituitary Stem Cells in Mammals'.

## Decision letter and Author response

Decision letter https://doi.org/10.7554/eLife.59142.sa1
Author response https://doi.org/10.7554/eLife.59142.sa2

## Additional files

### Supplementary files

• Supplementary file 1. Gene lists of gene set enrichment analyses. Gene lists generated from gene set enrichment analyses of bulk RNA-sequencing data comparing *Sox2*$^+$ and *Sox2*$^-$ cells. Associated with *Figure 3—figure supplement 1*.

• Transparent reporting form

### Data availability

Sequencing data can be accessed through the following link: https://www.ncbi.nlm.nih.gov/bioproject/PRJNA421806.

The following dataset was generated:

| Author(s) | Year | Dataset title | Dataset URL | Database and Identifier |
|---|---|---|---|---|
| Russell JP, Yianni V, Andoniadou CL | 2020 | Pituitary stem cells produce paracrine WNT signals to control the expansion of their descendant progenitor cells | https://www.ncbi.nlm.nih.gov/bioproject/PRJNA421806 | NCBI BioProject, PRJNA421806 |

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
