## [Decision Letter]

**Acceptance summary:**

This study uses a number of sophisticated genetic models to test the novel hypothesis that the pituitary stem cells produce WNT as a paracrine signal to drive proliferation of neighboring cells. The strengths of this study are that the authors use both in vitro and in vivo models to robustly test their hypothesis in a highly quantitative fashion. This study is of relevance to understanding development, disease, regeneration, aging and cancer of the pituitary gland, as well as stem cells, paracrine signalling and regeneration in other glandular and non-glandular organs.

**Decision letter after peer review:**

Thank you for submitting your article "Pituitary stem cells produce paracrine WNT signals to control the expansion of their descendant progenitor cells" for consideration by *eLife*. Your article has been reviewed by three peer reviewers, and the evaluation has been overseen by Marianne Bronner as the Senior and Reviewing Editor. The following individuals involved in review of your submission have agreed to reveal their identity: Elaine Emmerson (Reviewer #1); Sally Camper (Reviewer #2); Lori T Raetzman (Reviewer #3).

The reviewers have discussed the reviews with one another and the Reviewing Editor has drafted this decision to help you prepare a revised submission.

Summary:

This is a well-designed study that addresses an important question regarding paracrine signalling in the adult mouse pituitary gland. It fills a critical gap in knowledge regarding pituitary progenitors and also answers a longstanding question as to the role of WNT signaling in pituitary development. It is well written and most of the conclusions are well supported by the data.

Essential revisions:

1) Stating n numbers throughout to be transparent.

2) A more thorough analysis of the *Axin2^CreERT2/+^; Ctnnb1^LOF/LOF^* loss of function with regard to proliferation (to also look at the effect of tamoxifen), cell death and lineage distribution of GH, PRL and TSH cells would be important (e.g. apoptosis; SOX2; PIT1, TPIT, SF-1).

3) If it is not possible to increase n numbers where low (e.g. Figure 2C where n=2) the implications of this for data interpretation and statistics (or lack thereof) should be addressed and discussed in the manuscript.

4) The major point to support that the secretion of WNT from SOX2 cells is the key factor would require looking at the number of stem cells to make sure they are not changed with the WIS deletion. If this can't be done, the authors should tone down the conclusion from this experiment.

5) Clarifying whether both male and female mice were used throughout the study and if not, which sexes were used for which experiments.

The complete reviews are included below for your reference.

Reviewer #1:

This original manuscript demonstrates that SOX2^+^ pituitary stem cells are a key source of WNT ligands in the adult mouse pituitary gland, and the production of WNT from these cells is essential for paracrine signalling to regulate neighbouring cell proliferation. The approach taken is highly appropriate to address the experimental question, the experiments that were performed are elegant and logical and the conclusions drawn accurately represent the data presented. Appropriate statistical tests have been used and the figures are well presented and provide all relevant data. I have the following comments:

1) I appreciate that breeding sufficient numbers of complex genetic models can often be challenging, but can the authors be confident of the statistical power of experiments where they had n=2 (such as Figure 2C)?

2) It may not be possible, especially under the current circumstances, but have the authors investigated changes in any other markers in the *Axin2^CreERT2/+^*; *Ctnnb1^LOF/LOF^* mice, for example cell death (e.g. apoptosis); SOX2; or any of the differentiated lineages (PIT1, TPIT, SF-1)? It would be really interesting to see what else changes in the gland in absence of β-catenin signalling.

Reviewer #2:

This is an outstanding manuscript that uses multiple genetically engineered mice, stem cell cultures, and both gain and loss of function models to define the role of WNT signaling from pituitary stem cells to committed hormone-producing cell lineage progenitors in regulation of proliferation. Paracrine WNT secretion from stem cells is essential for the stimulation of proliferation in progenitors that have activated expression of lineage-specific transcription factors but not yet begun to produce hormones. This is an important step in understanding the molecular basis for craniopharyngiomas, which can be caused by gain of function mutations in β catenin, which cause non-cell autonomous acceleration of proliferation in neighboring cells. As such, it is an important advance for the field.

The paper is well-written and scholarly, drawing comparisons between the role of WNTs in pituitary development to the roles in other organ systems. For this reason, the work should be of broad interest for developmental biologists.

The authors conduct lineage tracing to determine which cells are responsive to WNT signaling using *Axin2^creERT2/+^* mice. The choice of postnatal day 14 for induction seems somewhat arbitrary. Is there any information about when WNT signaling is active during development? This may not change the conclusions, but it would be quite interesting to know when it is active. The timing may also bear on finding whether SF1 cells are significantly induced. Is there an expansion of gonadotrope cells around puberty? If so, then inducing at P14 and harvesting at P28, as puberty is commencing, may not be enough time to reach significance for SF1 cells. The authors noted that morbidity necessitated a limitation in the length of time permitted for analysis.

The authors focus on three lineage specific transcription factors: PIT1, TPIT and SF1. Since PIT1 specifies three hormone producing lineages (GH, PRL, and TSH), it could be worthwhile to assess whether there are any differential effects on the final differentiation into the three different cell types.

The authors demonstrate that the most WNT-responsive SOX2 expressing cells have the highest clonogenic potential, and by RNA sequencing of sorted cells they define the WNT ligands and frizzled receptors that the stem cells express. This is a valuable dataset as it also reveals expression of genes in other signaling pathways including Hippo, EGFR, ephrin, Mapk, FGF and p53. Noting the expression of *Wls*, which is necessary for WNT secretion, the authors deleted it in stem cells and discovered a reduction in proliferating progenitors, clearly establishing the need for WNT secretion in order to expand the stem cell compartment.

Can the authors clarify why a large proportion of cells appear to be tdTomato+ after 72 hrs when it appears in Figure 1D that ~30% are positive after 7 days?

The data presented are quite clear, quantified, and support the conclusions drawn from the data.

Reviewer #3:

This study is an important examination of the role of WNT signaling in postnatal pituitary expansion. The authors have a novel hypothesis that the pituitary stem cells produce WNT as a paracrine signal to drive proliferation of neighboring cells. In fact, paracrine signaling is the main role of the stem cells as opposed to being the primary source of new cells in the postnatal pituitary. It is a bit complicated that the WNT responsive, *Axin2* positive cells include both the stem cells and the lineage restricted cells. Additionally, WNTs are made in both stem and lineage restricted cells. This makes it hard to definitively say that WNT released from stem cells impacts the proliferation of the other lineage restricted cells. The experiment deleting *Wls* from SOX2 positive stem cells was an important way to address the role of paracrine WNT signaling. More details on this experiment are necessary to fully support the assertion that paracrine WNT signaling is an important driver of lineage restricted cell expansion.

1) For the *Sox2*-*Wls* conditional deletion experiment, there seems to be far fewer SOX2^+^ cells in the anterior lobe that in the control, based on the images in Figure 4A and B. If so, is it the loss of SOX2^+^ cells that leads to the reduced proliferation of the lineage committed cells and not the loss of WNT from the SOX2 cells? Confirming the number of SOX2 cells and contrasting this result to studies in which SOX2 cells are ablated at this time point would help to address this.

There are some additional comments to help clarify the manuscript:

2) The *axin2* conditional knockout experiments were only briefly described and the data presented in a supplementary figure. This could be highlighted more in that it shows WNT signaling is necessary for postnatal pituitary expansion. Although outside the scope of what is presented here, a pituitary specific temporally controlled *Ctnnb1* knockout would have been helpful to look more long term at postnatal loss of *Ctnnb1*.

3) At the end of the subsection “WNT/β-catenin signalling is required for long-term anterior pituitary expansion from *Sox2*^+^ pituitary stem cells”, it may be better to say the conclusion from those experiments is that WNT signaling is required in SOX2 positive stem cells to promote expansion of all pituitary populations.

4) The PTU-induced hypothyroidism experiment seems out of context with the story. *Axin2* cells aren't analyzed in detail. Although presented as a supplementary figure, it might be better to remove this data.

And a few experimental questions:

5) Does the tamoxifen administration alter proliferation in the pituitary? For example, could that have influenced the large expansion in GFP positive cells between PND2 and 7 in the experiments depicted in Figure 1. A control experiment showing proliferation during that time period with and without tamoxifen would help to allay this concern.

6) Why did you switch from analysis at p14 for the experiments in Figure 1 to p21 for the experiments in the first part of Figure 2? Are there any fundamental differences in the pituitary expansion at these ages?

7) For the *Ctnnb1* loss of function in *Sox2* cells, did you look after 5 days like was done for the *Ctnnb1* loss of function in all *axin2* positive cells? This may have helped compare directly what the contribution of *Ctnnb1* in stem cells is to the overall role of *Ctnnb1* in pituitary expansion at that age.

---

## [Author Response]

Essential revisions:1) Stating n numbers throughout to be transparent.

We have ensured that n numbers are stated throughout the manuscript.

2) A more thorough analysis of the Axin2^CreERT2/+^; Ctnnb1^LOF/LOF^ loss of function with regard to proliferation (to also look at the effect of tamoxifen), cell death and lineage distribution of GH, PRL and TSH cells would be important (e.g. apoptosis; SOX2; PIT1, TPIT, SF-1).

We have investigated the effect of tamoxifen on anterior pituitary proliferation in the *Axin2^CreERT2/+^* model and do not find any differences between control and tamoxifen-injected animals. Please see our response to point 5 of reviewer 3 and Author response image 1.

We have more thoroughly analysed the *Axin2^CreERT2/+^;Ctnnb1^LOF/LOF^* model as requested. We carried out cell death analysis by immunofluorescence against cleaved caspase-3, as well as analyses of the three committed lineages (by staining against PIT1, SF1 and ACTH) and SOX2 on 2 mutants and 3 control pituitaries. We have not observed significant differences in any of these analyses. The new data are provided as Figure 1—figure supplement 2B-D.

3) If it is not possible to increase n numbers where low (e.g. Figure 2C where n=2) the implications of this for data interpretation and statistics (or lack thereof) should be addressed and discussed in the manuscript.

Thank you for giving us the possibility to discuss the implications of the low numbers for the data interpretation and statistics. Unfortunately we have not been able to generate any further homozygous *Sox2^CreERT2/+^;Ctnnb1^LOF/LOF^* (or *Axin2^CreERT2/+^;Ctnnb1^LOF/LOF^* for new supplementary figures) animals during the pandemic in order to increase sample sizes. Upon lockdown the colonies had to be reduced down to the minimum and our *Ctnnb1-LOF* colony has suffered consequences, where we are trying to rescue this line. We hope that the reviewers are understanding of this situation. We have highlighted the limitations in the subsection “WNT/β-catenin signalling is required for long-term anterior pituitary expansion from SOX2^+^ pituitary stem cells”.

4) The major point to support that the secretion of WNT from SOX2 cells is the key factor would require looking at the number of stem cells to make sure they are not changed with the WIS deletion. If this can't be done, the authors should tone down the conclusion from this experiment.

We have analysed the stem cell compartment following *Wls* deletion. The number of SOX2 cells are reduced in *Sox2^CreERT2/+^;Wls^fl/fl^*mutants compared to *Wls^fl/fl^* controls (see below). These data support that WNT activation is also required by the SOX2^+^ cell compartment, a minority of which are also shown to be WNT responsive. We have included these new findings in the Results and refer to them in the Discussion.

*Sox2^CreERT2/+^;Wls^fl/fl^*mutants: 19.166% SOX2^+^/Total cells

*Wls^fl/fl^* controls: 23.425% SOX2^+^/Total cells

*P* = 0.0238, Student’s *t*-test, 5 mutants, 4 controls, over 2000 SOX2^+^ cells counted per genotype.

5) Clarifying whether both male and female mice were used throughout the study and if not, which sexes were used for which experiments.

Both male and female mice have been used throughout the study and the data pooled throughout. We do not expect sexual dimorphism before puberty (beginning around P26). Inductions have been carried out prior to this age. We have clarified this in the Materials and methods.

The complete reviews are included below for your reference.Reviewer #1:[…] I have the following comments:1) I appreciate that breeding sufficient numbers of complex genetic models can often be challenging, but can the authors be confident of the statistical power of experiments where they had n=2 (such as Figure 2C)?

We see an overwhelming reduction in dividing cells throughout the gland following long-term deletion of *Ctnnb1* in SOX2^+^ cells and descendants (n=2), accompanied by pituitary hypoplasia. We do agree with the reviewer that the statistical power of this would be low, and given that we have not been able to increase the sample number, we have indicated the sample size limitations in the subsection “WNT/β-catenin signalling is required for long-term anterior pituitary expansion from SOX2+ pituitary stem cells”.

2) It may not be possible, especially under the current circumstances, but have the authors investigated changes in any other markers in the Axin2^CreERT2/+^; Ctnnb1^LOF/LOF^ mice, for example cell death (e.g. apoptosis); SOX2; or any of the differentiated lineages (PIT1, TPIT, SF-1)? It would be really interesting to see what else changes in the gland in absence of β-catenin signalling.

Thank you for this suggestion. As per our response to the Editor’s summary comment 2., we have carried out the requested analyses in existing samples to look at apoptosis, SOX2, PIT1, TPIT and SF1 staining by immunofluorescence. We did not observe differences between controls and mutants. The new data are provided as Figure 1—figure supplement 2B-D.

Reviewer #2:[…] The authors conduct lineage tracing to determine which cells are responsive to WNT signaling using Axin2^creERT2/+^ mice. The choice of postnatal day 14 for induction seems somewhat arbitrary. Is there any information about when WNT signaling is active during development? This may not change the conclusions, but it would be quite interesting to know when it is active. The timing may also bear on finding whether SF1 cells are significantly induced. Is there an expansion of gonadotrope cells around puberty? If so, then inducing at P14 and harvesting at P28, as puberty is commencing, may not be enough time to reach significance for SF1 cells. The authors noted that morbidity necessitated a limitation in the length of time permitted for analysis.

There is a robust WNT activation in Rathke’s pouch from 11.5dpc. Thank you to the reviewer for bringing up this insightful point. The timeline on Figure 1B does indeed end at P28, before full commencement of puberty, which misses the expansion of SF1^+^ cells during puberty. We have therefore analysed the expansion of targeted WNT-responsive SF1^+^ cells at 28 days following induction at P14 (i.e. at P42), which spans puberty. This reveals a significant expansion of WNT-responsive SF1^+^ cells. The new data are included as Figure 1—figure supplement 1B and described in the subsection “WNT-responsive cells in the pituitary include progenitors driving major postnatal expansion”.

The authors focus on three lineage specific transcription factors: PIT1, TPIT and SF1. Since PIT1 specifies three hormone producing lineages (GH, PRL, and TSH), it could be worthwhile to assess whether there are any differential effects on the final differentiation into the three different cell types.

Thank you for the suggestion. We have analysed expansion of the three PIT1-derivatives and these data are presented as new Figure 1—figure supplement 1C. There is a preferential expansion of somatotrophs and thyrotrophs but not of lactotrophs. The data are described in the subsection “WNT-responsive cells in the pituitary include progenitors driving major postnatal expansion”.

The authors demonstrate that the most WNT-responsive SOX2 expressing cells have the highest clonogenic potential, and by RNA sequencing of sorted cells they define the WNT ligands and frizzled receptors that the stem cells express. This is a valuable dataset as it also reveals expression of genes in other signaling pathways including Hippo, EGFR, ephrin, Mapk, FGF and p53. Noting the expression of Wls, which is necessary for WNT secretion, the authors deleted it in stem cells and discovered a reduction in proliferating progenitors, clearly establishing the need for WNT secretion in order to expand the stem cell compartment.Can the authors clarify why a large proportion of cells appear to be tdTomato+ after 72 hrs when it appears in Figure 1D that ~30% are positive after 7 days?

The tdTomato reporter more closely resembles *Axin2* mRNA expression as seen by RNAScope. This is likely due to increased sensitivity of the *R26-tdTomato* reporter, recombining in more cells with lower *Axin2* expression than the *R26-mTmG* reporter. Additionally, the dosage of tamoxifen in Figure 2A (tdTomato) is double the amount used in Figure 1C (mTmG). For clarity, we have added a timeline for the mTmG experiments in Figure 1—figure supplement 1A.

The data presented are quite clear, quantified, and support the conclusions drawn from the data.Reviewer #3:This study is an important examination of the role of WNT signaling in postnatal pituitary expansion. The authors have a novel hypothesis that the pituitary stem cells produce WNT as a paracrine signal to drive proliferation of neighboring cells. In fact, paracrine signaling is the main role of the stem cells as opposed to being the primary source of new cells in the postnatal pituitary. It is a bit complicated that the WNT responsive, Axin2 positive cells include both the stem cells and the lineage restricted cells. Additionally, WNTs are made in both stem and lineage restricted cells. This makes it hard to definitively say that WNT released from stem cells impacts the proliferation of the other lineage restricted cells. The experiment deleting Wls from SOX2 positive stem cells was an important way to address the role of paracrine WNT signaling. More details on this experiment are necessary to fully support the assertion that paracrine WNT signaling is an important driver of lineage restricted cell expansion.1) For the Sox2-Wls conditional deletion experiment, there seems to be far fewer SOX2^+^ cells in the anterior lobe that in the control, based on the images in Figure 4A and B. If so, is it the loss of SOX2^+^ cells that leads to the reduced proliferation of the lineage committed cells and not the loss of WNT from the SOX2 cells? Confirming the number of SOX2 cells and contrasting this result to studies in which SOX2 cells are ablated at this time point would help to address this.

We show that SOX2 cells proliferate less in the absence of secreted WNTs (2.225% in mutants compared to 5.582% in controls). In addition based on the reviewer’s suggestion we show that they are fewer in number (19.166% in mutants compared with 23.425% in controls, see point 4 in response to the Editor’s summary, results in the subsection “Paracrine signalling from SOX2+ stem cells promotes WNT activation”). However, cycling SOX2^+^ cells are still present, at higher numbers than in other studies (Zhu et al., 2015, SOX2 cell reduction; Roose et al., 2017, SOX2 cell ablation). Even so, following *Wls* deletion, the reduction in surrounding cycling cells in the mutant is overwhelming, after only 7 days, where this cannot be explained by a reduction in proliferation of direct descendants of mutant SOX2 cells. Our confetti traces suggest that over a longer period, the majority of traced *Sox2* cells only appear to undergo one or two divisions (Figure 1—figure supplement 1D). This argues that the reduced proliferation we see is not due to commitment and expansion of SOX2 cells to a proliferative progenitor.

We do support the view that the mild reduction in SOX2 cells may be additive to the phenotype stemming from loss of WNT secretion following the deletion of *Wls*, reducing WNT levels even further. In the Zhu et al., 2015 study, where SOX2 cell numbers are reduced in *Rbp-J* mutants, proliferation of surrounding cells is also reduced. This points to a paracrine effect, especially given the low contribution of SOX2 cells to homeostasis. This hypothesis is discussed in the Discussion.

There are some additional comments to help clarify the manuscript:2) The axin2 conditional knockout experiments were only briefly described and the data presented in a supplementary figure. This could be highlighted more in that it shows WNT signaling is necessary for postnatal pituitary expansion. Although outside the scope of what is presented here, a pituitary specific temporally controlled Ctnnb1 knockout would have been helpful to look more long term at postnatal loss of Ctnnb1.

Thank you for the suggestion. We did not place further emphasis on the *Axin2*-driven *Ctnnb1* deletions, since the analysis is short-term and limited to 5 days post-induction due to morbidity. We believe that the key deletions of *Ctnnb1* in *Sox2*-expressing cells and their descendants illustrate this point with clarity (Figure 2C). Carried out over 22 weeks, the near-complete loss of Ki-67 positive cells and evident pituitary hypoplasia show that b-catenin is essential for pituitary expansion. This is consolidated in Figure 4, which reveals that without WNT signals from SOX2^+^ cells there is a reduction in cycling cells throughout the anterior pituitary.

At present there are no drivers to temporally control the *Ctnnb1* deletion only in the pituitary. *Hesx1^Cre/+^;Ctnnb1^LOF/-^* mice do not survive and neither do *Hesx1^Cre/+^;Wls^fl/fl^*, which was our alternative approach. We do not know of any inducible drivers solely targeting the pituitary postnatally or any loss of function mutations in *Ctnnb1* on reversible systems e.g. TetOn.

3) At the end of the subsection “WNT/β-catenin signalling is required for long-term anterior pituitary expansion from SOX2^+^ pituitary stem cells”, it may be better to say the conclusion from those experiments is that WNT signaling is required in SOX2 positive stem cells to promote expansion of all pituitary populations.

We have changed this sentence to: “In conclusion, WNT/b-catenin signalling is required cell-autonomously in SOX2^+^ stem cells and their descendants (Figure 2E).”

4) The PTU-induced hypothyroidism experiment seems out of context with the story. Axin2 cells aren't analyzed in detail. Although presented as a supplementary figure, it might be better to remove this data.

Upon the reviewer’s recommendation, we have removed these data on elevation of *Axin2* expression following PTU-induced hypothyroidism.

And a few experimental questions:5) Does the tamoxifen administration alter proliferation in the pituitary? For example, could that have influenced the large expansion in GFP positive cells between PND2 and 7 in the experiments depicted in Figure 1. A control experiment showing proliferation during that time period with and without tamoxifen would help to allay this concern.

From numerous past studies at the doses administered, tamoxifen administration has not been show to alter proliferation in the pituitary. To allay any concerns of the reviewer we carried out a control experiment on the *Axin2^CreERT2/+^;ROSA26^mTmG/+^* model and provide the data in Author response image 1. We do not see any difference in cycling cells marked by Ki-67, between corn oil or tamoxifen-injected pituitaries at P14, analysed 48 hours later.

6) Why did you switch from analysis at p14 for the experiments in Figure 1 to p21 for the experiments in the first part of Figure 2? Are there any fundamental differences in the pituitary expansion at these ages?

We do not detect any fundamental differences in clonogenic capacity at between P17 (P14+3 days) and P24 (P21+3 days). As the percentage of double positive (tdTomato^+^;EGFP^+^) cells is low, we opted for the experiments starting at P21 due to the larger pituitaries. This ensured we had sufficient numbers of double positive cells per experiment to increase the accuracy of colony counting.

7) For the Ctnnb1 loss of function in Sox2 cells, did you look after 5 days like was done for the Ctnnb1 loss of function in all axin2 positive cells? This may have helped compare directly what the contribution of Ctnnb1 in stem cells is to the overall role of Ctnnb1 in pituitary expansion at that age.

We did not look after 5 days as we tried to maximise the tracing periods whilst animals remain healthy. We agree with the reviewer that this would help a direct comparison and would be worthwhile to pursue in future studies.